



Manuscript prepared for Geosci. Model Dev.
with version 2015/04/24 7.83 Copernicus papers of the LaTeX class copernicus.cls.
Date: 3 May 2016

# Process-based modelling of the methane balance in periglacial landscapes with JSBACH

S. Kaiser[1], C. Beer[2], M. Göckede[1], K. Castro-Morales[1], C. Knoblauch[5], A. Ekici[3], T. Kleinen[4], S. Zubrzycki[5], T. Sachs[6], and C. Wille[5]

[1]Max Planck Institute for Biogeochemistry, Jena, Germany
[2]Stockholm University, Stockholm, Sweden
[3]University of Exeter, Exeter, United Kingdom
[4]Max Planck Institute for Meteorology, Hamburg, Germany
[5]Universität Hamburg, Hamburg, Germany
[6]Helmholtz Centre Potsdam GFZ German Research Centre for Geosciences, Potsdam, Germany

*Correspondence to:* S. Kaiser (skaiser@bgc-jena.mpg.de)

**Abstract.** A consistent, process-based methane module for a global land surface scheme has been developed which is general enough to be applied in permafrost regions as well as wetlands outside permafrost areas. Methane production, oxidation and transport by ebullition, diffusion and plants are represented. Oxygen has been explicitly incorporated in diffusion, transport by plants and two

oxidation processes, of which one uses soil oxygen, while the other uses oxygen that is available via roots. Permafrost and wetland soils show special behaviour, such as variable soil pore space due to freezing and thawing or water table depths due to changing soil water content. This has been integrated directly into the methane-related processes. A detailed application at the polygonal tundra site Samoylov, Lena delta, Russia, is used for evaluation purposes. The application at Samoylov also

shows differences in the importance of the several transport processes and in the methane dynamics under varying soil moisture, ice and temperature conditions during different seasons and on different microsites. These microsites are the elevated moist polygonal rim and the depressed wet polygonal center. The evaluation shows sufficiently good agreement with field observations despite the fact that the module has not been specifically calibrated to these data.

## 1  Introduction

Knowledge on atmospheric methane concentrations is a key factor in several global scale environmental research fields. Besides acting as a highly potent greenhouse gas and thus influencing global climate change, methane also contributes to degrading the ozone layer. Its average atmospheric lifetime is about 12.4 years, and its current atmospheric concentration in the Arctic is about 1850 ppbV

(Ito and Inatomi, 2012). Concentrations have been reported to rise slowly but steadily since the onset of industrialisation, and, after a hiatus at the beginning of the 21st century, have recently be found to rise again. These recent dynamics in the global atmospheric methane budget are still not fully



explained, emphasising the fact that also future trajectories of methane and its role in global climate change are highly uncertain. The global warming potential of methane is 84 to 86 times that of carbon dioxide over an integration period of 20 years and 28 to 34 times over 100 years (Myhre et al., 2013). Accordingly, even though its absolute mixing ratios are quite low compared to carbon dioxide, it makes up for about 20 % of the radiative forcing from all greenhouse gases. Thus, for the radiation balance and the chemistry of the atmosphere, it is important to understand land–atmosphere exchanges of methane.

Environmental conditions are highly heterogeneous in permafrost regions, where landscapes are often characterised by small-scale mosaics of wet and dry surfaces. The heterogeneous aerobic and anaerobic conditions in permafrost soils, in concert with elevated soil carbon stocks (Hugelius et al., 2014), set the conditions for large and spatially heterogeneous methane emissions in these areas (Schneider et al., 2009). Such strongly varying environmental and soil conditions as well as processes that influence the methane production and emissions are challenges in a process-oriented model with a bottom-up approach for methane balance estimation. However, process-based modelling approaches are powerful tools that help to quantify recent and future methane fluxes at large spatial scale and over long time periods in such remote areas. They can give first estimates where field measurements are missing and help to understand the effects of climate change on permafrost methane emissions. In addition, the effect of methane emissions on climate, hence feedback mechanisms, can be analysed using an Earth system model. For such purposes, a methane module for an Earth system model has to be process-based and working under most environmental conditions, including permafrost.

Currently existing process-based methane models have been usually developed for applications in temperate or tropical wetlands, without considering permafrost-specific biogeophysical processes, such as e.g. freezing and thawing soil processes, (e.g. Zhu et al., 2014; Schuldt et al., 2013). In other cases, they are embedded within a vegetation model, which cannot easily be coupled to an atmospheric model, (e.g. Schaefer et al., 2011; Wania et al., 2010; Zhuang et al., 2004). Some models have been developed only for small-scale applications (e.g. Xu et al., 2015; Mi et al., 2014; Khvorostyanov et al., 2008; Walter and Heimann, 2000) or use an empirical approach (e.g. Riley et al., 2011). Highly simplified models might be less reliable for global applications (e.g. Jansson and Karlberg, 2011; Christensen et al., 1996) because of oversimplification in simulating the complexity of the methane processes.

The aim of this study is to introduce a new methane module that is running as part of a land surface scheme of an Earth system model. Moreover, it shall be general enough for global applications, including terrestrial permafrost ecosystems. Therefore, the methane module presented in this work



represents the gas production, oxidation and relevant transport processes in a process-based fashion. Among other processes, this new methane module takes into account the size variation of the pore spaces in the soil column in relation to the freezing and thawing cycles, influencing directly the methane concentration in the soil. Furthermore, the oxygen content is explicitly taken into account, enabling a process-based description of oxidation processes.


The land surface scheme JSBACH (Jena Scheme for Biosphere Atmosphere Coupling in Hamburg) of the MPI-ESM (Max Planck Institute Earth System Model) was chosen for this work. The starting point was a model version that has a carbon balance (Reick et al., 2013), a five layer hydrology (Hagemann and Stacke, 2015) and includes permafrost as described in Ekici et al. (2014). A paral-

lel development by Schuldt et al. (2013) incorporated wetland carbon cycle dynamics and was also integrated in the model version presented in this work. The basis for the methane-related processes were the works by Walter and Heimann (2000) and Wania et al. (2010). Special focus was also put on the connections with permafrost and wetland as well as the explicit consideration of oxygen. This paper describes the newly developed methane module, and for the purpose of model evaluation it

presents an application at a typical polygonal tundra site in Yakutia.

## 2 Methods

### 2.1 Site description

For the purpose of evaluation, this model has been applied at the site Samoylov Island, located 120 km south of the Arctic Ocean in the Lena River Delta in Yakutia with an elevation of 10 to 16 m

above sea level. The mesorelief of Samoylov Island is flat, while as microrelief, there are low-center polygons with the soil surface about 0.5 m higher at the rim than at the center. This results in different hydrological conditions, also influencing heat conduction. The average maximum active layer depth at the dryer but still moist polygonal rims and the wet polygonal centers is at about 0.5 m (Boike et al., 2013). While the water table at the polygonal rims is generally well below the soil

surface, the polygonal centers are often water saturated with water tables at or above the soil surface (Sachs et al., 2008).

The vegetation on Samoylov Island can be classified as wet polygonal tundra that is composed of mosses, lichens and vascular plants. According to Kutzbach et al. (2004), mosses and lichens grow

about 5 cm high and cover about 95 %, while vascular plants grow about 20 to 30 cm high and cover about 30 % of the area. The most dominant vascular plant, both at the rim and at the center, is *Carex aquatilis* but with dominance of only 8 % at the rim compared to 25 % at the center. However, most of the species present at the rim are different from those present at the center. According to Sachs et al. (2010), the proportions of moist and wet microsites are approximately 65 % moist and 35 %



wet. The reader is referred to Sachs et al. (2010) for more details on the study site. Below, moist microsites will be referred to as rim and wet microsites as center.

## 2.2 Methane module description

### 2.2.1 Layer structure

For a numerically stable representation of gas transport processes in soils, a much finer vertical soil
structure is required than what is normally used for thermal and hydrological processes in JSBACH. Therefore, a new soil layering scheme has been implemented for the methane module. This scheme is variable and allows fine layers (in the order of a few $cm$) but still inherits the hydrological and thermal information contained in the coarse scheme. Number and height of layers can be chosen arbitrarily, allowing also non-equidistant solutions.


Internally, the module uses midpoints and lower boundaries of the layers as well as distances between the midpoints. At the bottom, the layering scheme is truncated at depth to bedrock. The layers, where the

- plant roots end, i.e., rooting depth lies,

- water table lies and

- minimum daily water table over the previous year lies (permanent saturated depth),

have also been determined. These layers have a specific function for methane production and various transport processes. Details will be given below in the respective sections (see also App. A1).

For model evaluation, fine layers with a height of 10 $cm$ have been used. For all the layers of the new soil layering scheme, the soil temperature is interpolated linearly from the coarse JSBACH layering scheme. From these values, also the previous day's mean soil temperature is calculated. In addition to geometry and soil temperature, each layer has its own hydrological parameters, as described in the next section, and various state variables describing the different gases' concentrations.

### 2.2.2 Hydrology

For the fine layers, several hydrological values have to be determined using the relative soil moisture and ice content from the coarse JSBACH layering scheme. Fine scale layer values are derived such that known values at common layers are kept and only those layers that span more than one input layer get values of the weighted mean of the involved coarse layer values. The relative soil water
content is then defined by the sum of the relative soil moisture and ice content.





Subtracting the relative ice content from the volumetric soil porosity leads to the ice-corrected volumetric soil porosity. With this, the relative moisture content of the ice free pores can be defined, which is calculated by division of the relative soil moisture content by the ice-corrected volumetric

soil porosity. Finally, the relative air content of the ice free pores is defined as one minus the relative moisture content of the ice free pores.

The water table is calculated following Stieglitz et al. (1997). From the uppermost soil layer, the water table is located in the immediate layer above the first one with a relative soil water content of

at least 90 % of field capacity. This definition was used because there is no oversaturation or standing water in JSBACH (Hagemann and Stacke, 2015). The dimensionless but ice-uncorrected field capacity is used because the relative soil water content already includes ice. The water table depth is then defined as

$$w = \begin{cases} b, & \text{if } r_w \leq 0.7 \cdot fc \\ b - \frac{r_w - 0.7 \cdot fc}{fc - 0.7 \cdot fc} \cdot h, & \text{if } r_w > 0.7 \cdot fc \, . \end{cases} \tag{1}$$

Here, $b$ is the lower boundary of the soil layer of interest with height $h$ and relative soil water content $r_w$. $fc$ is the field capacity. If even the uppermost layer has a relative soil water content of at least 90 % field capacity, the water table is located at the surface. The mean water table of the previous day is used where appropriate to keep consistency with the daily time step of the carbon decomposition routine. The minimum of this daily mean water table over the previous 365 days is used as the per-

manently saturated depth.

At a given time step, the soil column, that contains the water table depth and the permanently saturated depth, is divided into three strata that are from the top:

- the unsaturated zone above the water table,

– the saturated zone below the water table (located above the annual minimum water table depth) and

- the permanently saturated zone (located below the annual minimum water table depth).

Evidently, this stratification is hydrological, while the layering scheme is purely numerical. Thus, each stratum may contain several soil layers. For carbon decomposition, the mean temperatures of

the previous day at the midpoints of these three strata are needed. These values are derived analogously to the temperatures in the fine layers by interpolating the mean temperatures of the previous day linearly.

With these three strata, carbon that may experience unsaturated conditions is split into an unsat-

urated and a saturated pool by the water table. In addition, a permanently saturated carbon pool is



defined by the permanently saturated depth. This scheme is similar to what Schuldt et al. (2013) proposed. Further details about the calculation of the carbon decomposition are given in App. A2.

### 2.2.3 Production

Initial values of methane and oxygen concentrations have been derived using reported gas concentrations in free air for oxygen and methane. For oxygen, the global mean value for 2012 is used (8.56 mol m$^{-3}$, http://cdiac.ornl.gov/tracegases.html). The value for methane is defined as the March 2012 value (77.06 µmol m$^{-3}$, http://agage.eas.gatech.edu/data.htm).

The initial gas concentrations in the soil profile are determined assuming equilibrium condition between free ambient air as well as the air and moisture in the soil pore space. Thus, Henry's law with the dimensionless Henry constant is applied. The dimensionless Henry constant is defined as the ratio of the concentration of gas in moisture to its concentration in air (Sander, 1999). The chosen temperature dependence values, which are $\mathrm{d}\left(\ln k_{H\,\mathrm{CH_4}}\right)\left(\mathrm{d}(T^{-1})\right)^{-1}$ = 1900 K and $\mathrm{d}\left(\ln k_{H\,\mathrm{O_2}}\right)\left(\mathrm{d}(T^{-1})\right)^{-1}$ = 1700 K, as well as the Henry constants at standard temperature, which are $k_{H\,\mathrm{CH_4}}^{25}$ = 0.0013 mol dm$^{-3}$ atm$^{-1}$ and $k_{H\,\mathrm{O_2}}^{25}$ = 0.0013 mol dm$^{-3}$ atm$^{-1}$, are all from Dean (1992).

The calculated initial values for methane and oxygen concentrations in the soil profile can be transformed into gas amounts and vice versa. During methane transport process calculation, concentration values are widely used. In between time steps, however, the volume of ice is recalculated and therefore the relative ice free pore volume changes. Thus, concentration values also change, but only the gas amounts stay constant. Therefore, at the beginning of each methane module execution, the total gas amounts that have been saved at the end of the previous time step are divided by the current relative ice free pore volume to recalculate the current concentration values.

The final products of the decomposition of soil carbon are carbon dioxide and methane. Depending on the soil hydrological conditions, carbon dioxide or methane are produced from the decomposing carbon pools that belong to the three strata described above. These decomposition results are distributed over fine-scale layers of the whole soil column. Because no direct vertical information about the amount of decomposing carbon is available, equal decomposition velocity in all layers of one stratum is assumed. Thus, once the decomposed amount of carbon per stratum is known, the decomposed amount of carbon per layer per stratum depends on the amount of available carbon in that layer only. And the carbon content in the soil layers for Samoylov has been prescribed from measurements by Zubrzycki et al. (2013), Harden et al. (2012) and Schirrmeister et al. (2011), taking local horizontal variations of polygonal ground (Sachs et al., 2010) into account (see App. A3).





The initial amount of carbon in the pools is obtained from the sum of carbon in each layer of the strata. In this case, the first and second stratum share one carbon pool which is split after calculation of the mean water table over the previous day. The amount of carbon per layer is divided by the amount of carbon per stratum. These weights are used for distributing the amounts of decomposed carbon from strata to layers. In addition, the share of initially produced carbon dioxide and methane is set assuming all decomposed carbon above the water table and half of it below the water table gets carbon dioxide,

$$c_{prod}^{CH_4} = 0.5 \cdot \frac{f_C}{\sum_{sl} f_C} \cdot \frac{C_s}{h \cdot v_p} \ . \tag{2}$$

Here, $sl$ means all layers in the stratum, and $C_s$ is the decomposed carbon in the stratum. $f_C$ is the soil carbon content of the layer with height $h$, and $v_p$ is the ice-corrected volumetric soil porosity. Mass conservation is done if the stratum is too small to get a layer assigned, so that the associated carbon is not neglected. The gas fluxes for methane and carbon dioxide are calculated via the sums of the respective amounts, and the produced gases are added to their respective pools in the layers.

### 2.2.4 Bulk soil methane oxidation

Only part of the oxygen in the soil is assumed to be available for methane oxidation. In layers above the mean water table over the previous day, available oxygen is reduced by the amount that corresponds to the amount of carbon dioxide which is produced by heterotrophic respiration but not more than 40 % of the total oxygen content. Additional 10 % of oxygen is assumed to be unavailable and also reduced. In layers below the water table, the amount of oxygen is reduced by 50 %. This approach is similar to Wania et al. (2010).

For methane oxidation itself, a Michaelis–Menten kinetics model is applied. The $Q_{10}$ temperature coefficient is similar to the one used by Walter and Heimann (2000) but with a reference temperature of 10 °C rather than the annual mean soil temperature. Reaction velocities of both, methane and oxygen, are taken into account by using an additional equivalent term with the concentration of oxygen and $K_m^{O_2} = 2 \ \mathrm{mol\,m^{-3}}$, which is chosen to be the average concentration of oxygen at the water table. Furthermore, methane and oxygen follow a prescribed stoichiometry,

$$c_{oxid}^{CH_4} = \min\left( V_{max} \cdot \frac{c^{CH_4}}{K_m^{CH_4} + c^{CH_4}} \cdot \frac{c^{O_2}}{K_m^{O_2} + c^{O_2}} \cdot Q_{10}^{\frac{T-10}{10}} \cdot \mathrm{d}t, \ 2 \cdot c^{O_2}, \ c^{CH_4} \right) \ . \tag{3}$$

$c$ denotes the concentration of oxygen or methane in the layer. $T$ is the soil temperature in the layer, and $\mathrm{d}t$ is the time step. The total gas fluxes for methane, oxygen and carbon dioxide are again calculated as the sums of the respective amounts.

### 2.2.5 Ebullition

The implementation of the ebullition of methane follows largely the scheme from Wania (2007). Ebullition is the transport of gas via bubbles that form in liquid water within the soil and transport





methane rapidly from their place of origin to the water table. The amount of methane to be released through ebullition is determined by that amount of the present methane that can be solute in the present liquid water. This amount depends on the overall amount of methane present in the layer but also on the storage capacity of the present liquid water.


In a first step, the concentration of methane in soil air is assumed to be in equilibrium with the concentration in soil water. Thus, by application of Henry's law, the present methane can be partitioned into the potentially ebullited methane concentration in soil air and the potentially solute methane concentration in soil water. The dimensionless Henry solubilities at current soil tempera-

ture conditions are used for this. As initial approximation, all methane is assumed to be in soil air and potentially ebullited. Thus, first, the potentially solute methane in soil water can be determined, but it will also be overestimated because of this approximation. Therefore, second, an updated potentially ebullited concentration of methane in soil air is determined by subtracting the potentially solute methane from the total methane. Unlike proposed in Wania (2007), these two steps are iterated

until stable state conditions are reached.

In a second step, to calculate the maximal amount of methane that can be soluble in the present soil water, the Bunsen solubility coefficient from Yamamoto et al. (1976) is applied. By considering the available pore volume, this gives the volume of methane that can maximally be dissolved. The

ideal gas law results in the maximally soluble amount of methane. For that, the soil water pressure in layers below the water table needs to be derived. This is determined from soil air pressure and the pressure of the water column, using the basic equation of hydrostatics. For this, the specific gas constant of moist air and the soil air pressure in layers above the water table are required. For the air pressure calculation, the barometric formula is used. Hereby, the first layer uses the air pressure

at the soil surface and deeper layers use the above layer's soil air pressure. The specific gas constant of moist air finally needs the saturation vapour pressure and relative soil air moisture, both in layers above the water table. The former is calculated after Sonntag and Heinze (1982), and the latter is set to 1 if the relative water content is at least at the wilting point and to 0.9 elsewhere.

Now, the maximally soluble concentration of methane is derived by dividing the maximally soluble amount of methane by the available pore volume. Thus, the concentration of methane that is solute and in equilibrium with methane in the air is the lesser of the following two concentrations: the potentially solute methane, that was calculated in the first step, and the maximally soluble methane, that was calculated in the second step. Finally, the actually ebullited methane is the difference between

all methane and solute methane,

$$c_{ebul}^{CH_4} = c^{CH_4} - \min\left( k_{H\,CH_4} \cdot c_{gas}^{CH_4}, \frac{\beta \cdot p_w}{R \cdot T} \right) , \qquad (4)$$



with $k_{H\,CH_4}$ being the Henry solubility, $c_{gas}^{CH_4}$ the methane concentration that can potentially be ebullited, $\beta$ the Bunsen solubility coefficient, $p_w$ the soil water pressure and $T$ the soil temperature, all of the layer, and $R$ is the gas constant.


The ebullited methane is removed from the layers and, if the water table is below the surface, added to the first layer above the water table. In this case, the ebullition flux to atmosphere is zero, and the methane is still subject to other transport or oxidation processes in the soil. Otherwise, if the water table is at the surface and if snow is not hindering, it is added to the flux to atmosphere. Snow is

assumed not to hinder if snow depth is less than 5 cm. If, finally, the water table is at the surface but snow is hindering, ebullited methane is put into the first layer and the ebullition flux to atmosphere is zero like in the first case.

### 2.2.6 Diffusion

For the diffusion of methane and oxygen, Fick's second law with variable diffusion coefficients

is applied. The possibility of a non-equidistant layering scheme is specifically taken into account. Diffusion is a molecular motion due to a concentration gradient, with a net flux from high to low concentrations. For soil as a porous medium, moreover with changing pore volumes because of different contents of ice, the ice-corrected soil porosity of the layers also has to be accounted for in the equation system directly as a factor (Schikora, 2012). The discretisation of the computational

system is done with the Crank–Nicholson scheme with weighted harmonic means for the diffusion coefficients. While ice is treated as non-permeable for gases, the diffusion is allowed to continue if the soil is frozen but not at field capacity, i.e., there is no simple cut at $0\,^{\circ}\mathrm{C}$. During every model time step of 1 hour, two half-hourly diffusion steps are calculated to prevent instabilities like oscillations or unrealistic behaviour like negative concentrations. The diffusion specific time step can

be decreased further if necessary and if an adjustment of the layering scheme is not desired. The possibility of these effects results from the tight connection between layering scheme, time step and diffusion coefficients.

As initial condition, free ambient air, soil air and moisture phase are assumed to be in equilibrium.

The boundary condition at the bottom of the soil column is always of Neumann type, i.e., no flux is assumed. At the top of the soil column, boundary conditions are assumed to depend on snow depth. If there are at least 5 cm of snow, no flux is assumed, and therefore Neumann type is applied also at the top. However, if there are less than 5 cm of snow, ambient air conditions are assumed to hold at the boundary, and therefore Dirichlet type with gas concentration in free air is applied,

$$\boldsymbol{v}_p \cdot \frac{\partial \boldsymbol{c}}{\partial t} = \frac{\partial}{\partial x}\left(\boldsymbol{D} \cdot \frac{\partial \boldsymbol{c}}{\partial x}\right) \; ; \quad c = c_{air} \, , \quad x \in \Gamma_D \; ; \quad \frac{\partial c}{\partial x} = 0 \, , \quad x \in \Gamma_N \, . \tag{5}$$





Here, $v_p$ is the volumetric soil porosity, $c$ denotes the gas concentration, $t$ is the time, $x$ is the depth, $D$ denotes the diffusion coefficient, $\Gamma_D$ is the boundary with Dirichlet type boundary conditions, and $\Gamma_N$ is the boundary with Neumann type boundary conditions. For details on how the diffusion coefficients are determined, see App. A4. The solution of the diffusion equation system is obtained

by the `tridag_ser` and `tridag_par` routines from Press et al. (1996) in Numerical Recipes.

By subtracting the gas concentrations after diffusion from those before for methane and vice versa for oxygen, concentration changes are derived with positive values for lost methane and gained oxygen. Multiplying the concentration changes with their respective pore volumes as usual and summing

the resulting amounts over the layers gives the total fluxes of methane and oxygen.

### 2.2.7 Plant transport

Gas transport via plants is first calculated for oxygen entering the soil. Then, another oxidation mechanism with this newly gained oxygen takes place (see Sect. 2.2.8). After that, the transport of methane via plants is simulated. The transport via plants happens through the plant tissue, that con-

tains big air filled channels, the aerenchyma, to foster aeration of the plant's roots. However, because plants need the oxygen that reaches their roots for themselves, their root exodermis acts as efficient barrier against gas exchange.

In this model configuration, gas transport is assumed to happen only via the phenology type grass

with C3 photosynthetic pathway. Furthermore, this transport occurs only if snow is not hindering, i.e., if there are less than 5 cm of snow. This is justified by the consideration of snow crinkling the culms such that transport is not possible anymore. A diffusion process from aerenchyma through the root tissue to soil is assumed as key process, and it is described by Fick's first law. Gas transport is fast inside the air-filled aerenchyma, hence, atmospheric air conditions can be assumed there.


The diffusion flux via the plants is determined from the oxygen concentration gradient between ambient air and the root zone soil layers. The diffusion coefficients of methane and oxygen in the exodermis are unknown but can be assumed to be slightly lower than in water (e.g. Kutzbach et al., 2004; Končalová, 1990). Therefore, their values are set to be 80 % of their respective values in soil

water at the given soil temperatures and pressures, $D_r = 0.8 \cdot D_w$.

The oxygen flux entering the soil is furthermore constrained by the surface area of root tissue, $A_r^{ges} = A_r \cdot q_p$, which is determined from the surface area of a single plant's roots, $A_r = l_r \cdot d_r \cdot \pi$, multiplied by plant density, $q_p = \frac{t_{ph}}{t_p}$. Here, $l_r$ is the root length, $d_r$ the root diameter, both in me-

tres, $t_{ph}$ the number of tillers per square metre depending on phenology, and $t_p$ the number of tillers per plant. Finally, the number of tillers per square metre is influenced by plant phenology, which is





determined from the $LAI$, using $t_{ph} = \max(t_m) \cdot \frac{LAI}{\max(LAI)}$, with $t_m$ being the number of tillers per square metre. Please see also App. A5.

The root tissue is assumed to be distributed equally between all root-containing layers, thus $A_r^{rl} = A_r^{ges} \cdot \frac{h}{\sum_{rl} h}$, with $h$ denoting the layer height and $rl$ all layers with roots. The travel distance, $\mathrm{d}x$, is set to the thickness of the exodermis in metres because this is the limiting factor. The plant transport per layer is thus modelled as

$$n_{plant}^{\mathrm{O_2}} = D_r^{\mathrm{O_2}} \cdot \left( c_{air}^{\mathrm{O_2}} - c^{\mathrm{O_2}} \right) \cdot \frac{1}{\mathrm{d}x} \cdot \mathrm{d}t \cdot A_r^{rl} \ . \tag{6}$$

Here, $c_{air}^{\mathrm{O_2}}$ is the concentration of oxygen in free air and $\mathrm{d}t$ the time step length. For every soil layer, the resulting amount of oxygen is converted into concentration and added to the oxygen pool. As usual, the flux of oxygen into the soil is calculated by the total soil column balance.

After plant transport of oxygen, additional methane can be oxidised by the amount of oxygen that
leaves the roots (Sect. 2.2.8). The remaining methane is then available for plant transport, which is modelled exactly as for oxygen, with one exception: It is necessary to account for the fraction of roots able to transport gases, $f_r = \frac{dom_{CarexA.}}{dom_{VascularP.}}$. This can be thought of as a measure of distance between the methane and the transporting roots. With increasing amounts of roots being able to transport gases, the distance for methane to travel to them is getting smaller and transport is gener-
ally enhanced. To account for that, $f_r$ is set for rim and center, respectively, as the fraction of the dominance measure for *Carex aquatilis* divided by the dominance of vascular plants (Kutzbach et al., 2004). The plant transport of methane is thus modelled as

$$n_{plant}^{\mathrm{CH_4}} = D_r^{\mathrm{CH_4}} \cdot \left( c^{\mathrm{CH_4}} - c_{air}^{\mathrm{CH_4}} \right) \cdot \frac{1}{\mathrm{d}x} \cdot \mathrm{d}t \cdot A_r^{rl} \cdot f_r \ . \tag{7}$$

The variables definitions are the same as for oxygen and $c_{air}^{\mathrm{CH_4}}$ is the concentration of methane in
free air. A similar effect will be taken into account for oxygen when it is allowed to oxidise only methane near the transporting roots. To determine the flux out of the soil, the differences of methane concentrations in the soil subtracted by the concentration in ambient air are used. For every layer, the amount of methane is converted into concentration and removed from the methane pool. Again, the total methane flux out of the soil is calculated by summing up individual layer balances.

### 2.2.8 Rhizospheric methane oxidation

The oxygen gained by the transport via plants is assumed to foster methane oxidation next to their roots. Thus, if oxygen is leaving these roots, the same oxidation routine as described above in Sect. 2.2.4 is applied to calculate how much additional methane is oxidised by this oxygen. Obviously, only gas concentrations in layers with roots will be influenced. Because the amount of vegetation
with roots that are able to supply oxygen varies between rim and center, the dominance measure ($f_r$





from Sect. 2.2.7) is applied again as a factor to account for the distance to these roots,

$$c_{plox}^{\mathrm{CH_4}} = \min\left( V_{max} \cdot \frac{f_r \cdot c^{\mathrm{CH_4}}}{K_m^{\mathrm{CH_4}} + f_r \cdot c^{\mathrm{CH_4}}} \cdot \frac{c_{plant}^{\mathrm{O_2}}}{K_m^{\mathrm{O_2}} + c_{plant}^{\mathrm{O_2}}} \cdot Q_{10}^{\frac{T-10}{10}} \cdot \mathrm{d}t, \ 2 \cdot c_{plant}^{\mathrm{O_2}}, \ f_r \cdot c^{\mathrm{CH_4}} \right) . \quad (8)$$

The variables' definitions are the same as for the bulk soil methane oxidation, $f_r$ is the fraction of roots in the layer that are able to transport gases, and $c_{plant}^{\mathrm{O_2}}$ is the concentration of oxygen trans-

ported by plants. Carbon and oxygen pools are adjusted accordingly. The total exchange with the atmosphere is determined by summing the total amount of gas that is calculated by multiplying the concentrations by their pore space.

## 2.3 Simulation setup

As a global land surface scheme, JSBACH does not represent lateral water flow, which, however, is

a process of major importance in polygonal tundra sites. To account for the different hydrological conditions at polygon rim and center, the performed experiment consisted of two different simulation runs with different settings for rim and center. The polygon rim is assumed to be a normal upland soil, and a standard JSBACH simulation run was performed. For center, runoff and drainage of the rim have been collected and added to center precipitation. Additionally, for the center run, runoff

and drainage have been switched off until the soil water content reached field capacity.

The sequence of methane processes executed in the module is identical to the above described order within Sect. 2.2.1 to 2.2.8, and has been sorted according to the velocity of the specific processes, from fast to slow. The impact of changing this sequence on total and component methane flux rates

was tested in a separate sensitivity study (not shown). These tests indicated only a minor influence of the sequence to the partitioning of the fluxes between the transport processes compared to the influence of hydrology or the definition of the processes themselves. Still, it cannot be excluded that simulated methane processes may be biased through the chosen order under certain conditions.

The carbon pools for rim and center were initialised using data from Zubrzycki et al. (2013) and information from Harden et al. (2012), Schirrmeister et al. (2011) and Sachs et al. (2010). The used values for rim and center for Samoylov are $627.61 \ \mathrm{mol\,m^{-2}}$ and $731.94 \ \mathrm{mol\,m^{-2}}$ for the upper carbon pool and $16355 \ \mathrm{mol\,m^{-2}}$ and $25424 \ \mathrm{mol\,m^{-2}}$ for the lower carbon pool. Because of the lack of information on how the modelled soil carbon from these two pools is distributed vertically,

a depth distribution is applied to the decomposed carbon instead. For all layers within one stratum, equal decomposition velocity is assumed. The relative amounts of measured carbon are applied as distribution aid for the decomposed carbon. The layers used were $10 \ \mathrm{cm}$ in height.

The only other parameters varying between rim and center are the number of tillers per square metre

and the dominance of *Carex aquatilis*, two vegetation parameters in the process of plant transport.



Otherwise, the model has not been calibrated to site specific processes or properties. The used grid cell size was 0.5 °.

To run the model, an initial hydrological spin-up has been done, using a mean climate year from
the period of observations that has been repeated 100 times. Only after 40 of these spin-up years, the methane processes have been switched on to give the hydrology the possibility to stabilise before the methane processes were allowed to take place. After 100 years of spin-up, the time period of interest has been calculated with actual climate data.

### 2.4  Forcing and evaluation data

The climate forcing data used in the simulations is the same as in Ekici et al. (2014), spanning from 14 July 2003 to 11 October 2005. The climate input consists of air temperature, precipitation, atmospheric relative humidity, short and long wave downward radiation and wind speed, all at hourly resolution.

For model evaluation, data from chamber measurements has been used. This data was collected over 39 days from July to September 2006 by Sachs et al. (2010), resulting in 55 single measurements for the rim and 48 for the center. In addition, eddy covariance based fluxes from Wille et al. (2008) have been used, integrating rim and center. From this, 3340 data points were available for the simulation time period.

## 3  Results

### 3.1  Modelled water table and permanent saturated depth

The modelled depth of permanent saturation for both, rim and center, is always at the same level of 31.9 cm. In contrast, the modelled water table changes during the seasons for rim and center differently (Fig. 1). In general, it is higher at the center than at the rim, though there are few cases in
early spring when the rim has a higher water table than the center. This results from the different soil water contents at the rim and at the center which were forced by adding runoff and drainage from the rim to the center as precipitation and prohibiting runoff and drainage at the center until the soil water content reached field capacity. Still, in the early part of the thawing season, the water tables at the rim and at the center are similar. While in general, at the rim, the water table is highest during the
early thawing season, at the center, there is a tendency to high values towards the end of the thawing season. But if the rim shows a high water table, there will generally also be a high water table at the center. Overall, the water table in the model is changing relatively quickly, due to the quick changes in modelled soil water conditions.



However, JSBACH does not allow for oversaturation or standing water. Thus, the maximal soil water content in the model is field capacity. It is obvious, that there is a mismatch with the real situation in the field, where the center is often water saturated with water tables at or above the soil surface. While measurements of the water table at the rim give values between 35 and 39 cm (Kutzbach et al., 2004), the mean summer value in the model is 30.88 cm. For the center, measurements give values

between -12 and 17 cm (Sachs et al., 2010), while the mean summer value in the model is 24.52 cm. Hence, the model tends to have a slightly higher water table at the rim, but the calculated water table is too low at the center. Still, this water table has been calculated using the unsaturated soil water content. That there is no oversaturation in the model and that therefore the soil moisture content is incomparable to the situation in the field should be kept in mind while interpreting the results of the

methane module.

For additional results concerning modelled physical conditions, such as soil moisture and ice content as well as soil temperatures, the reader is referred to App. B1 to B3.

### 3.2 Modelled methane flux in summer and winter

The modelled methane fluxes at rim and center are different for the different seasons (Fig. 2). While most of the modelled flux is positive (i.e. emission to the atmosphere), there are also uptake events. The spread of the flux is greater for the center than for the rim in both summer and winter. While the majority of flux values in summer is positive at the center, it is more balanced at the rim. In winter, the methane flux is almost always zero, following the assumption that snow may hinder the exchange.

However, at the center, there are some rare events when uptake takes place. In the mixed approach, which means 65 % rim and 35 % center, the overall mean emission is about 0.0813 $\mathrm{mgC\,m^{-2}\,h^{-1}}$ for the summer period. The overall higher emissions at the center are due to higher moisture and thus more favourable conditions for methane production in concert with lower methane oxidation rates.

### 3.3 Cumulative sums

During most parts of the year, the diffusive methane flux is rather small at the rim (Fig. 3A) and sometimes slightly negative at the center (Fig. 3B). During spring, however, there are large methane bursts happening. They are fed by the methane that accumulated in the soil during winter and that is released as soon as the snow melts. Plant mediated methane transport is smaller than diffusion but more pronounced at the center than at the rim (Fig. 3A and 3B) because plant transport was

defined to be slower than diffusion in water and should thus lead to lower emissions under less wet conditions. However, the wetter the soil, the more plant transport relative to diffusion should occur, because the more water the more is diffusion slowed down. While ebullition is the most important process at the center (Fig. 3B), it is diffusion at the rim (Fig. 3A). This is due to the drier conditions at the rim that allow a fast diffusion through air, while ebullition is only possible with a minimum of





soil moisture. Because in the model, higher soil moisture is calculated from the middle to the end of the thawing season, most of the emissions by ebullition and plant transport at the center occur then (Fig. 3B).

In the mixed approach, only the diffusion of the rim alters the pattern of the emissions substan-
tially (Fig. 3C). In total, the polygon center accounts for a 6.8 times as large fraction of emissions as the rim due to the higher methane production under wetter conditions (Fig. 3D). This means, a total share of 78.6 % of the methane emissions in the mixed approach is coming from the center. Emissions at the rim are highest during spring, while they are highest at the center during the mid and late season (Fig. 3D).


When comparing the total fluxes of the center to the ones of the rim, diffusion is almost doubled, plant transport is 19 times as high, and ebullition is 18 times as high (Table 1). This results in almost seven times higher total methane emissions at the center than at the rim. While diffusion at the rim is more than 13 times as high as plant transport at the rim, the diffusion at the center is just slightly
higher than the plant transport there. Ebullition is about 4.5 times as high as plant transport both at the rim and at the center. These differences are again due to the differences in soil moisture content, which allow more production under higher soil moisture and thus also lead to more methane emissions. On the other hand, plant transport is in principle a slower transport process than diffusion in water, but diffusion in water is much slower than diffusion in air. Thus, under drier conditions, diffu-
sion in air will transport the main portion, but under wetter conditions, plant transport may increase relative to diffusion. With reduced soil air, the remaining velocity of the diffusion is almost at the same order of magnitude like the overall velocity of plant transport, in contrast to the velocity of diffusion mainly through air.

Still it seems, that the plant transport in the model is too low compared to the total flux. While the diffusion flux to the atmosphere only happens at the soil surface, the surface area of the gas trans-porting roots is the relevant boundary for plant transport. The value of this is not well-known, so the module might need further adjusting of parameters connected to plant root surface area to improve the share of plant transport. Furthermore, ebullition needs substantial amounts of soil moisture, and
this is more common at the center than at the rim. Consequently, substantially more ebullition is found at the center than at the rim. In the mixed approach, diffusion accounts for about 2.5 times of the emissions of plant transport, while ebullition accounts for 4.5 times of it. Overall, 0.588 g of carbon are emitted by each square metre during the modelled time period from 14 July 2003 to 11 October 2005.



### 3.4 Split into the different transport processes

Splitting the total methane flux into several transport processes shows differences in the amount of their contribution per process, depending on the rim or center position, but also differences in the pattern in time (Fig. 4A). In general, the fluxes are much lower at the rim than at the center (Fig. 4B) because it is drier there and less methane is produced. Ebullition adds large portions to the total balance at both microsites at isolated time steps, reflecting the nature of this process, while its total amount for rim is rather small. At the rim, diffusion represents both the second largest methane release and substantial uptake during the season (Fig. 4A). The smallest flux portion at the rim is due to plant transport, which also shows some uptake. In contrast, plant transport plays a much more pronounced role at the center. Diffusion shows more negative than positive fluxes there. In spring, methane produced during winter and stored under the snow gets released as large bursts both at the rim and at the center. All these effects occur in the different hydrological regimes at the rim and at the center. Based on the assumption that plant transport is slower than diffusion in water, the resulting pattern of flux processes and soil moisture were expected. Still it seems, that the plant transport is too low compared to the total flux, which is subject to further investigations. Oxygen available to consume methane plays another modulating role, in particular for plant transport.

### 3.5 Production versus oxidation

Methane oxidation follows the pattern of methane production as long as enough oxygen is available (Fig. 5A). Production, and hence also oxidation, is higher during times of more moist conditions for both, the rim and the center, and also higher for the center than for the rim (Fig. 5B). At the center, a substantial amount of methane is oxidised in the rhizosphere with oxygen that enters the soil via plant transport. This happens when a high amount of methane is produced, which is rather rare at the rim due to lower soil moisture (Fig. 5A). During spring, bursts of oxidation occur both at the rim and at the center because methane produced during the winter and stored below the snow gets in contact with oxygen. The different moisture and temperature regimes at the rim and the center and their dynamics determine these results.

### 3.6 Comparison to chamber measurements

Although the number of available field data is small and from a different year than the meteorological forcing data, the field measurements and model results are of the same order of magnitude (Fig. 6). Observations and model results show higher center values compared to the rim, but the model seems to underestimate occasional uptake events. For the rim, the model gives methane fluxes to the atmosphere between -0.0237 and 39.3 $\mathrm{mgC\,m^{-2}\,h^{-1}}$ with mean 0.0267 $\mathrm{mgC\,m^{-2}\,h^{-1}}$, while the available field measurement values range from -0.111 to 0.881 $\mathrm{mgC\,m^{-2}\,h^{-1}}$ with mean 0.154 $\mathrm{mgC\,m^{-2}\,h^{-1}}$. For the center, the model gives values between -0.0189 and 86.8 $\mathrm{mgC\,m^{-2}\,h^{-1}}$ with





mean 0.231 $\mathrm{mgC\,m^{-2}\,h^{-1}}$, while the available field measurement values range from -0.0584 to 1.22

$\mathrm{mgC\,m^{-2}\,h^{-1}}$ with mean 0.327 $\mathrm{mgC\,m^{-2}\,h^{-1}}$. Besides higher mean values, the extremes are thus lower for the field measurements. This is due to the observation period excluding spring time when the model calculates the highest emissions (spring bursts). Those are the result of an accumulation of methane below the snow during winter.

One should also take into account that JSBACH is a global model, therefore it requires input parameters from global fields. Furthermore, other modules of JSBACH, like the hydrology or the carbon decomposition, are adjusted for global applications. Therefore, JSBACH integrates processes over much larger grid cell areas than what chamber measurements may represent. Hydrological conditions and other processes are highly variable in polygonal tundra environments and are of crucial

importance for methane processes. Still, they may not be represented with the required detail by the model so that the modelled conditions are the same as those at the measurement site. Thus, it is obvious, that with coarser and different hydrological conditions, the modelled methane fluxes per square metre for a 0.5 ° grid cell cannot be identical to the point measurements of chambers. Particularly, the low soil moisture in the hydrological conditions of the model may explain the lower mean

modelled methane fluxes compared to what is reported by chamber data.

### 3.7 Comparison to eddy measurements

Eddy covariance data had the best available data coverage of field measurements (light grey areas in Fig. 7). Overall model results are of the same order of magnitude as observations, but there are also seasonal shifts between model results and measurements. This is due to a mismatch between

the real soil conditions at the measurement site and the modelled soil climate and hydrology, that cannot be expected to be the same as those in the field. The range of available measurements in the modelled period is 0.0233 to 4.59 $\mathrm{mgC\,m^{-2}\,h^{-1}}$ with mean 0.609 $\mathrm{mgC\,m^{-2}\,h^{-1}}$. The range of modelled summer methane emissions in this time frame is -0.023 to 30.4 $\mathrm{mgC\,m^{-2}\,h^{-1}}$ with mean 0.0813 $\mathrm{mgC\,m^{-2}\,h^{-1}}$. If less than 5 cm of snow are on the ground, this is defined as summer time.

Besides lower mean values, the model shows higher extremes.

For this comparison, the same constraints hold like for the comparison to chamber data. The modelled fluxes have to differ from the field measurements because of the less moist modelled hydrological conditions. Modelled periods with no emissions at the same time as substantial emissions in

the field measurements show, that also the combination of temperature and hydrology of the model does not always fit the conditions at the field site. During these time periods, there is a lot of soil ice in the model and temperatures are well below zero. Thus it is natural, that there are no emissions.

Still, Fig. 7 also shows some patterns that are present in both model results and observations, e.g.





periods with increasing fluxes that are followed by a sudden decline in the fluxes in a cyclic manner during a single season. This patterns are linked to the changing soil moisture content. Unfortunately only the first season is covered well by field measurements, while the second misses the later part, and the third covers just a part within. Moreover, there are no measurements available for spring time when the model shows the highest methane emissions (spring bursts), which are the result of

an accumulation of methane below the snow during winter.

For additional results concerning modelled oxygen uptake, such as mixed daily sum, seasonally split and cumulative sums as well as transport process split, see App. B4.

## 4 Discussion

This paper aims to present the methods of a new methane module for the land surface scheme JS-BACH. Its purpose is to show how the module works in principle and with concrete data. The module itself is completely integrated into the model JSBACH. Thus, it is not possible to examine the performance of the module separately from the rest of the model. All conclusions to be made should therefore consider the distinction between the module functioning and the JSBACH model perfor-

mance as a whole.

The presented methane module determines production, oxidation and transport of methane to the atmosphere. In order to do that, among others it depends very much on soil hydrology and carbon decomposition, which both are handled by other JSBACH modules. If these modules are constrained

to lack some features that would be relevant for the methane processes just because the final scope is to use JSBACH globally, the methane module may not be expected to represent site level data without limitations. This fact is even more important if taking into account, that JSBACH also uses a lot of parameters from global fields that are naturally not that exact as if they were all measured at the same field site where the fluxes were measured. It is almost obvious, that with that many systematic

deviations also the model results may be systematically different from the site level measurements. Thus, it may be more informative to compare the methane fluxes to the modelled hydrological conditions and amounts of decomposed carbon instead of to site level data.

Still, the comparison of site level measurements are an important step but also controversial in

evaluating a process-oriented global biosphere model. In particular, the limited amount of available field measurements from chamber and eddy covariance based fluxes requires a careful interpretation when compared to model results. The question of representativeness always arises in the temporal domain because of the discontinuous type of the methane fluxes (e.g. Tokida et al., 2007; Jackowicz-Korczyński et al., 2010; Tagesson et al., 2012) and in the spatial domain because of the



large variability of soil organic matter and water content. Therefore, long-term and widespread flux measurements are highly important for any model evaluation.

Even so, comparisons to scarce field data are helpful, and the comparison to eddy covariance based fluxes is treated as the most reliable. But it also has to be considered, that under many conditions,
the footprint composition of the eddy covariance tower might not match the mixed approach of 65 % rim and 35 % center used for modelling (Sachs et al., 2010). This is an approximation to cope with the hydrological constraints of a global model on the one hand and the complex landscape on the other. Particularly when footprints are smaller during daytime, the field data might focus on areas that are wetter or drier than the average.


Moreover, the modelled soil climate and hydrology are not the same as those to be found in the field. The model hydrology, e.g., is a global one that has not been designed to be applied in such complex landscapes like polygonal tundra for site level detailed analysis. Still, we used this partic-ular site because overall data coverage was good. To adapt the model to the complex hydrology, a
mixed approach of combining two different model runs was applied. This is a very much simplified hydrological approach compared to reality and still cannot offset all site level differences between model and reality. Therefore, hydrological details are not the same as in the field. However, using this approach, it was found, how critical a reasonably fitting hydrology for the methane balance is. For example, many details in the behaviour of the methane processes follow strictly the varying hy-
drological conditions during the year or between the microsites.

The model application for remote permafrost areas may also be limited by the availability of long-term and complete observations of meteorological data to be used as model forcing. Forcing data and methane fluxes are required for the same time period, which optimally lasts over one or more years.
When going towards regional to global applications, this new model might be additionally compared to regional or global atmospheric inversion results (e.g. Bousquet et al., 2011; Berchet et al., 2015) or data-oriented upscaling of eddy covariance or chamber based observations (e.g. Christensen et al., 1995; Marushchak et al., 2015).

Within the methane module presented in this work, the discretisation as well as the pore volume are variable, thus the time step of calculation and the diffusion coefficients must fit to the thicknesses of the soil layers. Otherwise, instabilities like oscillations or unrealistic behaviour like negative con-centrations may occur. This module has been designed flexible in this respect, and adjustments can easily be made.


Furthermore, assumptions, e.g. about winter fluxes or plant transport, might be too strict accord-



ing to newer findings (Zona et al., 2016; Marushchak et al., 2015). The prohibition of gas exchange with the atmosphere under conditions with more than 5 cm of snow on the ground is an adaption to the modelled hydrological conditions in winter. Because of too little soil water, these conditions were

allowing unreasonably high methane uptake during winter time. On the other hand, this approach has the tradeoff, that reasonable exchange with the atmosphere during winter is also prohibited.

The definition of the plant transport follows a mechanistic approach with weak knowledge about velocities and parameters in reality. Arising from published statements, the value for the diffusion

coefficient in the exodermis was chosen to be 80 % of the diffusion coefficient in water. However, gas transport within the aerenchyma is assumed to be as quick as diffusion in air. Still, if the barrier of the root exodermis is effective, transport will be limited by this barrier. The geometric size of this barrier has a large influence on plant transport, too. While a thinner root exodermis would lead to more plant transport, it is relatively easy to determine the thickness of a root exodermis. However,

the cumulative surface area of all gas transporting roots in the soil column is not at all easy to determine. But the larger this surface, the larger the plant transport. If it is found, that plant transport is too low compared to the other transport pathways, it is likely that also the chosen value for the surface area of gas transporting roots is not yet optimal. These kind of issues are the subject of ongoing investigations, but the module has been designed flexible, and adjusting of parameters with respect

to newer findings is easily possible.

In Samoylov Island, the minimum of modelled daily sums of methane emissions during summer is smaller and the maximum much higher for rim and center compared to measurements published by Kutzbach et al. (2004). However, these observations do not include spring bursts with very short

but also very high emissions or even dry phases with small uptake. On the other hand, the mean of those measurements is 3 times as high for rim and 3.5 times as high for center compared to the modelled daily sums in summer (Table 2), high modelled emissions are rather rare, and the general level of modelled values is lower than in observations (Fig. 7).

When comparing our model results at Samoylov Island to published results from other high-latitude regions, reasonable agreement is found. Our modelling results are about 40 to 60 % lower than measurements for BOREAS, Canada, and Abisko, Sweden, (Wania et al., 2010). Samoylov is much colder and drier which suggests reasonably lower fluxes. Compared to measurements done by Desyatkin et al. (2009) on a thermokarst terrain at the Lena river near Yakutsk, our mean results are well

within the measurement range if comparing our rim to the drier sites, our center to the wetter sites, and our mixed approach to the entire ecosystem (Table 3). But climate and environmental conditions are different from those in Samoylov, so more than a rough overview comparison has no value. Nakano et al. (2000) measured methane fluxes at Tiksi near the mouth of the Lena river. While our



mean value at rim is 4.5 times as high as the mean measurements in Tiksi, the mean at the center

is 5.5 times as high as our mean value (Table 2). The modelled minimum is lower for the center but comparable for the rim. Furthermore, much higher maxima have been determined for both rim and center most probably due to the earlier described events of spring bursts, that have not been measured. In Zona et al. (2009), several measurements of methane emissions in the Arctic tundra are given. Despite our mean values are located towards the lower end, our minimum, mean and

maximum values fit well within the given range, that shows a widespread of possible observations. Bartlett et al. (1992) measured methane fluxes near Bethel in the Yukon–Kuskokwim delta, Alaska. The provided values for upland tundra compare well to our mean and minimum values. However, the model maximum fluxes are higher than the measurement values for upland tundra but still well in the range of measured values for wet meadow, which has higher moisture contents than upland

tundra. In fact, the highest values are calculated if soil moisture is highest, so despite more on the lower end of this water logged landscape type's emissions, they fit well also therein. Even so, all these places have different conditions in terms of weather, carbon pools and so on. Thus, despite the modelled values are within a plausible range at the greater picture, more information cannot be gained here.


Still, other parts of the land surface scheme require advancements before applying it with the methane module at global scale and over long time periods can be suggested. For example, soil organic matter should be represented vertically resolved, with different soil carbon pools and a moisture dependent decomposition (e.g. Braakhekke et al., 2011, 2014; Koven et al., 2015). Furthermore, the site hydrol-

ogy should include oversaturation and standing water (Stacke and Hagemann, 2012). Calculating a water table depth empirical after Stieglitz et al. (1997) from the unsaturated soil water content is obviously not the best solution, despite the fact, that not only the water table depth but also the soil moisture content itself is of great importance to the presented methane module. However, already with the presented model version, the importance of different processes, their interplay or the influ-

ence of climatic or hydrologic drivers can be studied at site level, which is a major step forward: This process-based implementation can be applied at other sites or with another hydrology, and still, the methane-related processes will only depend on the soil conditions. Despite being a complex process model, the interplay of the processes is consistent. Thus, the influence of climate and hydrology on methane fluxes can be studied in detail. Which process is most important under which condi-

tions gives useful information about the complex behaviour of the methane dynamics in permafrost soils. In sum, a lot of information can be gained from using this model that all may help understand the complex network of drivers, influencing factors and constraints that govern methane balance in periglacial landscapes.



## 5 Conclusions

The aim of this study was to develop a consistent, process-based methane module for a land surface scheme which is also reliable in permafrost ecosystems. Based on previous work by Wania et al. (2010) and Walter and Heimann (2000), the land surface scheme JSBACH of the global Earth system model MPI-ESM has been enhanced for this purpose. The new methane module represents methane production, oxidation and transport. Methane transport has been represented via ebullition, diffu-

sion and plant transport. Oxygen can be transported via diffusion and plant transport. There are two oxidation pathways, one takes the amount of soil oxygen into account and the other uses explicitly oxygen that is available via roots. All methane-related processes respond to different environmental conditions in their specific ways. They increase or decrease according to their requirements with changing soil moisture, temperature or ice content. The differences between the processes, seasonal

differences as well as differences between the microsites rim and center have been shown.

When combined with a module for oversaturated soil conditions like TOPMODEL (e.g. Kleinen et al., 2012), such methane-advanced land surface scheme can be used to estimate the global methane land fluxes, including for periglacial landscapes. These regions are rich in soil carbon (Hugelius et

al., 2014) and show good conditions for methane production (Schneider et al., 2009). However, they are often remote and rather hard to investigate. Thus, process-oriented modelling can contribute to understand the role of methane emissions as long as widespread and long-term measurements remain scarce. In addition, the role of methane for future permafrost carbon feedbacks to climate change can be studied. For these reasons, the module in this study is highly integrated also with permafrost and

wetland processes, e.g., changing pore space in the soil because of freezing and thawing or changing water table depths due to changing soil water content. In a first comparison with site level field measurements, sufficiently good agreements could be shown, despite the module has not been adjusted to site specific processes or features. Coupling such land surface scheme to atmosphere and ocean schemes in an Earth system model will provide the basis for studying methane-related feedback

mechanisms to climate change.

## 6 Authors' contributions

S. K. developed the model code and performed the simulations. S. K. also designed the whole study and prepared the manuscript together with C. B., M. G., K. C.-M. and C. K.. Permafrost module code and input data have been provided by A. E., and T. K. provided code for the parallel developed

wetland representation. S. Z., T. S. and C. W. provided field observations.



## 7  Code availability

The model code used in this work is available upon request for academic and non-commercial use.

## Appendix A:  Additional methods

### A1  Layer structure – specific layer determination

Specific layers are determined by comparing the midpoints of the layers to rooting depth, water table or minimum daily water table over the previous year, respectively. If one of these lies between two layer midpoints, the layer with the upper midpoint is chosen to be the specific layer for that. If the depth under consideration and the midpoint of a layer are the same, the corresponding layer is chosen.

### 765  A2  Hydrology – decomposition of carbon

The decomposition of carbon is determined similar to Schuldt et al. (2013) though appropriate temperatures are used for each of the three strata. Furthermore, the decomposition times for the three carbon pools have been adjusted to ensure that the two pools under partially oxic conditions are relatively stable, neither accumulating nor decomposing great portions within a few years, and the
last pool slowly accumulating. In numbers, the former two pools change only about $1\,\mathrm{mol\,m^{-2}}$ each within the calculation period from 14 July 2003 to 11 October 2005. The decomposition time scales used are 80, 400 and 30000 years for the unsaturated, currently saturated and permanently saturated stratum's carbon pool.

Though the rate of organic matter decomposition at the evaluation site is not known, the present-day amount of carbon in the soil is known (Sect. 2.2.3). Considering short time scales only, the above described approach should give reasonable amounts of decomposed carbon in the three strata. This way, the input to our methane routine, the amount of decomposed carbon per time step in each stratum, is provided daily.

### 780  A3  Production – soil carbon per layer

The amount of soil carbon per layer has been prescribed based on measurements for the first metre of the soil profile by Zubrzycki et al. (2013). The values of the six measurement depths were averaged over the sixteen different center respectively six rim cores. These resulting averages have been interpolated to $1\,\mathrm{cm}$ values for rim and center accordingly. The means of the corresponding $1\,\mathrm{cm}$
values are then used for the modelling layers within the first metre of the soil profile.

As Zubrzycki et al. (2013) only give values for the first metre, additional information for the rest





of the soil profile is needed. Schirrmeister et al. (2011) give an estimate for Lena delta soil carbon content of 553.33 $\mathrm{kg\,m^{-2}}$ with a soil depth of 18.25 m, which is converted in a volumetric estimate of 30.32 $\mathrm{kg\,m^{-3}}$. Harden et al. (2012) give quantitative information about the depth distribution of soil carbon up to 3 m. Horizontal variations are accounted for by a partitioning in 65 % rim and 35 % center (e.g. Sachs et al., 2010).

Using this information, values are assigned to the remaining layers so that the overall mean over all layers, rim and center mixed in the proposed partitioning, gives the volumetric estimate gained from Schirrmeister et al. (2011). Hereby, the information from Harden et al. (2012) about the variability over depth, that is a slight decrease up to 1.7 m and a slight increase thereafter, is taken into account.

As uppermost values for this, at a depth of 1.05 m, the mean of the deepest measured values are taken as 21.24 $\mathrm{kg\,m^{-3}}$ for rim and 35.00 $\mathrm{kg\,m^{-3}}$ for center. As values at the turning point, in depths of 1.65 to 1.75 m, the ceiled mean values of the first metre are used, which are 20 $\mathrm{kg\,m^{-3}}$ for rim and 34 $\mathrm{kg\,m^{-3}}$ for center. In between, the values are interpolated, towards the depth extrapolated linearly to meet the criterion of overall fitting to the value of Schirrmeister et al. (2011) as mentioned above.

### A4 Diffusion – diffusion coefficients

After Collin and Rasmuson (1988), the diffusion coefficients of methane and oxygen in the soil layers are calculated by adding the diffusion coefficients in soil moisture times the dimensionless Henry solubility to the diffusion coefficients in soil air. Both are weighted by the relative pore moisture respective air content, and the ice-corrected soil porosity of the modelling layers is also considered. The exponents for this are estimated with Newton's method. For fast convergence, an appropriate starting value has been chosen, that was found to be 0.62. The dimensionless Henry solubilities for methane and oxygen at the current soil temperatures are applied, and the diffusion coefficients in soil air and moisture are derived.

The diffusion coefficients in soil air can be seen as such in free air at soil temperature and pressure. They are calculated after Massman (1998) from values at the soil surface with over depth variable soil temperature and pressure. The latter one arises from soil air and water pressure. The values of diffusion coefficients in free air at soil surface are calculated from values at 0 °C and 1 atm (Massman, 1998).

The diffusion coefficients in soil moisture can be seen as such in free water at soil temperature and pressure. They are calculated differently for the two gas species. For methane, Jähne et al. (1987) is




used, whereas for oxygen, Boudreau (1996) is used with the calculation of the dynamic viscosity of water after Matthaus as quoted by Kukulka et al. (1987),

$$D = \left(1 - \frac{r_m}{v_p}\right)^2 \cdot (v_p - r_m)^{2 \cdot \epsilon_a} \cdot D^a_{(0,1)} \cdot \left(\frac{T}{T_0}\right)^{1.81} \cdot \frac{p_1}{p_s} + k_H \cdot \left(\frac{r_m}{v_p}\right)^2 \cdot r_m^{2 \cdot \epsilon_w} \cdot D^w . \tag{A1}$$

Here, $r_m$ is the relative soil moisture content, $v_p$ the ice-corrected volumetric soil porosity, $\epsilon_a$ and $\epsilon_w$ the exponents from Collin and Rasmuson (1988) for air and water, $T$ the soil temperature, $p_s$ the soil air respective water pressure in $\mathrm{atm}$ and $k_H$ the Henry constant, all of the layer. $D^a_{(0,1)}$ is the diffusion coefficient in free air at $T_0 = 273.15\mathrm{K}$ and standard pressure $p_1 = 1$ atm, and $D^w$ is the diffusion coefficient in water under the conditions of the layer. The latter two for methane and oxygen are defined as

$$\begin{aligned}
D^a_{\mathrm{CH_4}\,(0,1)} &= 1.952 \cdot 10^{-5} \ \mathrm{m^2\,s^{-1}} , & D^w_{\mathrm{CH_4}} &= A \cdot exp\left(-\frac{E_a}{R \cdot T}\right) , \\
D^a_{\mathrm{O_2}\,(0,1)} &= 1.820 \cdot 10^{-5} \ \mathrm{m^2\,s^{-1}} , & D^w_{\mathrm{O_2}} &= \left(0.2604 + 0.006383 \cdot \frac{T}{\mu}\right) \cdot 10^{-9} \ \mathrm{m^2\,s^{-1}} .
\end{aligned} \tag{A2}$$

with $A$ and $E_a$ from Jähne et al. (1987), and $R$ being the gas constant. $T$ is once more the temperature and $\mu$ the dynamic viscosity of water, both of the layer.

To establish the boundary conditions for the system properly, for both the upper and lower boundary of the soil column one additional computational point has to be added to the computational system. Also for the boundary conditions, but just for computational reasons, two virtual points in the same distance from the upper respective lower boundary as the first respective last inner point are needed. These points have as properties their location and diffusion coefficient only, which are the same as those of the first respective last layer. The layer heights are used as weights for the weighted harmonic means of the diffusion coefficients at the borders between the layers. Just if boundary points are involved, half of the layer heights are used as weights.

### A5 Plant transport – setup details

The thickness of the exodermis is set to 0.06 $\mathrm{mm}$ (Kutzbach et al., 2004). The number of tillers per square metre for rim and center are given by Kutzbach et al. (2004). The number of tillers per plant is set to one. While the mean accumulated root length of one plant is derived from Shaver and Billings (1975) to be 0.739 $\mathrm{m}$, the root diameter is derived from Kutzbach et al. (2004) to be 1.9 $\mathrm{mm}$.

### Appendix B: Additional results

#### B1 Modelled relative soil moisture content

The modelled soil moisture content changes seasonally very much. However, because soil water content is restricted to field capacity, there is also a limit for soil moisture content at field capacity.





At the rim (Fig. 8A), soil moisture increases in the upper soil part in spring but decreases with
ongoing thawing season. In contrast, at the center (Fig. 8B), soil moisture increases only slowly in
spring, but this increase is ongoing until almost the end of the thawing season. This is due to the
greater amount of ice in the soil, which thaws slowly. On the other hand, the greater input of water
to the center than to the rim as soon as there is runoff created at the rim is a continuous additional
supply of soil moisture to the center later in the thawing season. With this, the rim is more moist than
the center in the beginning of the thawing season but drier in the middle and at the end of it (Fig.
8C). Just in the deeper layers, rim has a little bit more liquid water during the whole thawing season.
In winter, however, the amount of liquid water is negligible both at the rim and at the center. Thus,
differences may only be seen in the timing of changes due to thawing respective freezing, which
both happen earlier at the rim than at the center. Consequently, they result in earlier wetting of the
rim's soil during spring as well as earlier drying of it during freezing.

**B2    Modelled relative soil ice content**

The modelled soil ice content, in contrast, is almost always higher at the center than at the rim.
Only during freezing in autumn, there is a short period when there is more ice in the uppermost
soil part at the rim than at the center. During the thawing season there generally is very little ice in
the upper part of the rim's soil (Fig. 9A), while at the center, small amounts of ice may also occur
in this period (Fig. 9B). Both, rim and center, show substantial amounts of ice below 30 cm even
during the summer. Furthermore, during spring, while the uppermost part of the soil at the center is
already thawed, an accumulation of new ice takes place right below, which thaws shortly after. In
general, the upper soil part gets its ice thawed and frozen more slowly and later at the center than at
the rim because there is more ice at the center. Below 30 cm, the difference in ice content between
rim and center is increasing in summer (Fig. 9C). However, this levels off during freezing, until it
reestablishes in winter at a lower level. In winter, the soil part with the least amount of ice is not on
top but between 10 and 30 cm both at rim and center.

**B3    Modelled soil temperature**

The modelled soil temperatures show deeper thawing and higher temperatures during the thawing
season at the rim compared to the center (Fig. 10A). In addition, rim temperatures reach lower
values in winter. Moreover, the thawing season starts earlier and ends later for the rim than for the
center (Fig. 10B). These effects are due to the generally drier soil at the rim compared to the center.
Water dampens the amplitude of the temperature change, and in addition, the phase change takes up
energy. While the warming to 0 °C occurs quickly, the phase change takes time and the soil can only
warm further after the phase change is completed. During freezing, the reverse occurs. The cooling
then is faster and to lower temperatures at the rim compared to the center. In general, deeper layers
react more slowly and dampened compared to layers close to the surface. At the rim as well as at





the center, there are short periods with temperatures below 0 °C even during summer. The highest
temperature differences occur during early spring when there is more ice in the ground at the center
than at the rim. Thus, the rim can reach the zero curtain easier (Fig. 10C).

### B4 Modelled oxygen uptake

#### B4.1 Mixed daily sum

The overall pattern of oxygen uptake shows big portions during the early and late thawing season
with a reduced uptake during the mid season (Fig. 11). This is the most moist part of the season,
and water effectively reduces oxygen diffusion into the soil. There is also some daily variation in
the amount of uptake during the thawing season, that is connected to the soil moisture content. The
wetter the soil, the less oxygen can enter. Because there is high uptake at the beginning and the end
of the thawing season, the overall transport of oxygen is more similar for the rim and the center, in
contrast to methane, where the center is dominating. In winter, no uptake takes place because snow
hinders the exchange.

#### B4.2 Seasonal split

The modelled oxygen uptake at the rim and at the center is different for the different seasons (Fig.
12). In summer, the uptake is purely positive and greater for the rim than for the center. Also, the
spread of uptake is greater for the rim than for the center. This is again due to the drier conditions
that allow more diffusion through air, which is quicker and can thus lead to higher uptake compared
to diffusion in water or via plants under the wetter conditions at the center. In winter, the uptake is
zero, following the assumption that snow hinders the exchange. In the mixed approach, the overall
mean uptake is about 2.21 $\mathrm{gO_2\,m^{-2}\,h^{-1}}$.

#### B4.3 Cumulative sums

At the rim, diffusion delivers a much greater portion of oxygen than plant transport (Fig. 13A). At
the center, both processes provide almost the same amount of oxygen (Fig. 13B). There are no such
pronounced bursts during spring as for methane. While plant transport is smaller than diffusion for
both, rim and center, the difference is much bigger at the rim. At the center, there is more plant
transport but less diffusion than at the rim. Diffusion at the rim and plant transport at the center
are increasing towards the end of the thawing season. In contrast, diffusion at the center and plant
transport at the rim show decreasing contributions towards the end of the thawing season.

In the mixed approach, rim and center add to a relatively uniform increase of oxygen flux by dif-
fusion over the whole thawing season. For plant transport, the mid season increase is highest, with
smaller contributions at the beginning and the end of the thawing season (Fig. 13C). This results




from the different timing of high soil moisture content at the rim and at the center that compensate each other for diffusion. Furthermore, the wetter the soil, the more plant transport relatively to diffusion should occur, because the more water the more is diffusion slowed down. If, moreover, these conditions occur towards the end of the growing season, which is the case at the center, the effect is bigger than if this happens in spring, which is the case at the rim. Still, diffusion accounts for a larger proportion of uptake than plant transport because plant transport was defined to be slower than diffusion in water while diffusion in air is rather quick. It might still be, that the plant transport is too low compared to the total uptake because the root surface might have been chosen too small, like the results for the methane emissions suggest. In total, the rim accounts for more oxygen uptake than the center (Fig. 13D), but the difference is not as high as for the methane emissions. While the late season is slightly more important at the rim, it is the early season for the center.

When comparing rim and center total uptake, diffusion gets reduced to about a third at the center compared to the rim, and plant transport gets almost 4 times as high (Table 4). This results in a reduction to less than two-thirds of the overall uptake at the center compared to the rim. While at the rim, diffusion is almost 12 times as high as plant transport, they are almost at the same level at the center. These differences are again due to the differences in soil moisture content. In the mixed approach, diffusion accounts for about 4.5 times of the uptake of plant transport. Overall, 16 kg of oxygen are taken up by each square metre in the course of the modelled time period.

**B4.4 Transport process split**

Splitting the overall oxygen uptake into the transport processes shows differences in the amount of their contribution per process, depending on location, but also differences in the pattern (Fig. 14A). The uptake is split into different portions between the processes, that are more equal for the center (Fig. 14B) but differ a lot for the rim. There, diffusion is responsible for the majority of the uptake. At the center, this is only true in the early season and at the freezing. In the mid season, plant transport is much higher than diffusion. While the diffusion part is lower at the center than at the rim, the opposite is the case for plant transport. In spring, big amounts of oxygen are taken up both at the rim and at the center. In the late season, also some small emissions via diffusion occur at the center. In general, uptake through diffusion is greater when soil is drier, which is the case for the rim in the late and for the center in the early season. While plant transport is more steady at the rim, there are pronounced peaks at the center when the soil is wettest. In spring, when the soil is wettest at the rim, plants are not yet that far developed that plant transport could increase to similarly high values as at the center during the respective times with high soil moisture content.

*Acknowledgements.* This work has been supported by the PAGE21 project, grant agreement number 282700, funded by the EC seventh Framework Programme theme FP7-ENV-2011, and the CARBOPERM project, grant agreement number 03G0836B, funded by the BMBF (German Ministry for Science and Education). Addi-



tional support came from the PerCCOM project, grant agreement number PCIG12-GA-2012-333796, funded
by the EC seventh Framework Programme theme FP7-PEOPLE-2012-CIG, and the AXA Research Fund,

PDOC_2012_W2 campaign, ARF fellowship M. Göckede.

Special thanks go to Uwe-Jens Görke for his interdisciplinary support and to Guido Krämer for his support
in graphical issues.



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





**Table 1.** Maximal cumulative methane flux.

|                | Rim    | Center | Mixed  |
| -------------- | ------ | ------ | ------ |
| Diffusion      | 0.139  | 0.268  | 0.182  |
| PlantTransport | 0.0103 | 0.196  | 0.0752 |
| Ebullition     | 0.0492 | 0.876  | 0.339  |
| All            | 0.194  | 1.32   | 0.588  |

Maximal values of the cumulative sums of modelled methane flux over the modelled time period for rim, center and a mixed approach of 65 % rim plus 35 % center for the different transport processes and combined in $\mathrm{gC\,m^{-2}}$, rounded to three non-zero digits.

**Table 2.** Summary of daily methane flux.

|        | Min    | Mean | Max |
| ------ | ------ | ---- | --- |
| Rim    | -0.690 | 1.34 | 208 |
| Center | -0.208 | 8.21 | 385 |
| Mixed  | -0.521 | 2.90 | 135 |

Modelled daily methane flux for the summer periods 2003 to 2005 for rim, center and a mixed approach of 65 % rim plus 35 % center in $\mathrm{mgCH_4\,m^{-2}\,d^{-1}}$, rounded to three non-zero digits. Summer means less than 5 cm snow are on the ground. Please note the different unit here.

**Table 3.** Summary of hourly methane flux.

|        | Min     | Mean   | Max  |
| ------ | ------- | ------ | ---- |
| Rim    | -0.0237 | 0.0267 | 39.3 |
| Center | -0.0189 | 0.231  | 86.8 |
| Mixed  | -0.0235 | 0.0813 | 30.4 |

Modelled hourly methane flux for the summer periods 2003 to 2005 for rim, center and a mixed approach of 65 % rim plus 35 % center in $\mathrm{mgC\,m^{-2}\,h^{-1}}$, rounded to three non-zero digits. Summer means less than 5 cm snow are on the ground.



**Table 4.** Maximal cumulative oxygen uptake.

|  | Rim | Center | Mixed |
| --- | --- | --- | --- |
| Diffusion | 17.0 | 5.97 | 13.2 |
| PlantTransport | 1.45 | 5.41 | 2.84 |
| All | 18.5 | 11.4 | 16.0 |

Maximal values of the cumulative sums of modelled oxygen uptake over the modelled time period for rim, center and a mixed approach of 65 % rim plus 35 % center for the different transport processes and combined in $\mathrm{kgO_2\ m^{-2}}$, rounded to three non-zero digits.



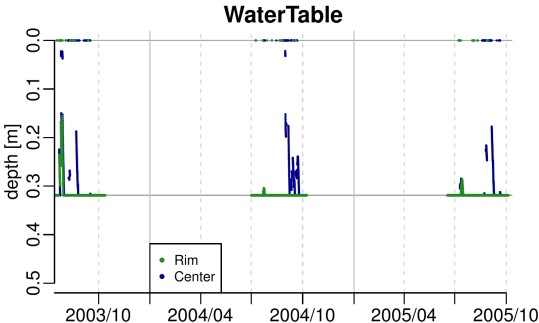

**Figure 1.** Modelled water table at rim and center. Solid lines indicate 1st of January, dashed lines indicate 1st of April, 1st of July and 1st of October of the respective year. Only the summer periods are shown, which means less than 5 cm snow are on the ground.

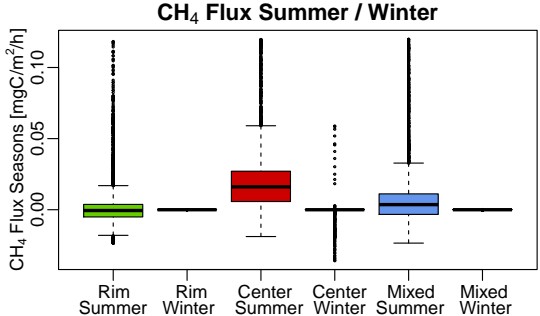

**Figure 2.** Modelled methane flux out of soil at rim, center and a mixed approach of 65 % rim plus 35 % center, split into summer and winter. Summer means less than 5 cm snow are on the ground, winter is the remainder. Because of the widespread of values, from -0.0747 $\mathrm{mgC\,m^{-2}\,h^{-1}}$ to as high as 86.8 $\mathrm{mgC\,m^{-2}\,h^{-1}}$, a portion of 4.66 % values was cut to provide a reasonable picture.





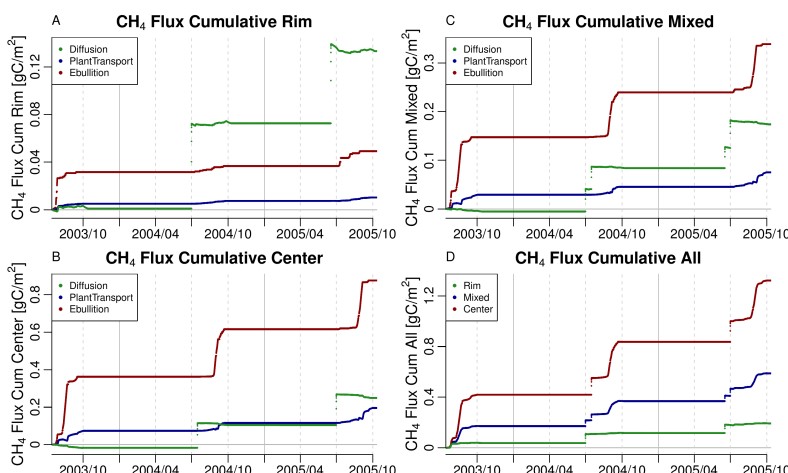

**Figure 3.** Modelled methane flux out of soil at (A) rim, (B) center, (C) a mixed approach of 65 % rim plus 35 % center, split into the different transport processes, and at (D) rim, center and a mixed approach of 65 % rim plus 35 % center, combined, as cumulative sum. Solid lines indicate 1st of January, dashed lines indicate 1st of April, 1st of July and 1st of October of the respective year. Please note the different scales. Table 1 gives the maximal values.



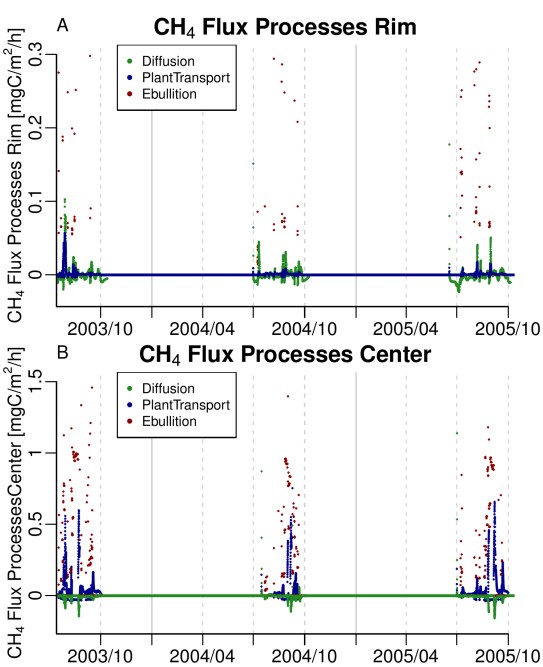

**Figure 4.** Modelled methane flux out of soil at (A) rim and (B) center, split into the different transport processes. Solid lines indicate 1st of January, dashed lines indicate 1st of April, 1st of July and 1st of October of the respective year. Please note the different scales. Because of the widespread of high values, to as high as 39.3 (A) and 86.6 (B) $\mathrm{mgC\,m^{-2}\,h^{-1}}$, a portion of 0.108 % (A) and 0.0609 % (B) values was cut to provide reasonable pictures. The minima of the values are -0.0234 (A) and -0.158 (B) $\mathrm{mgC\,m^{-2}\,h^{-1}}$.





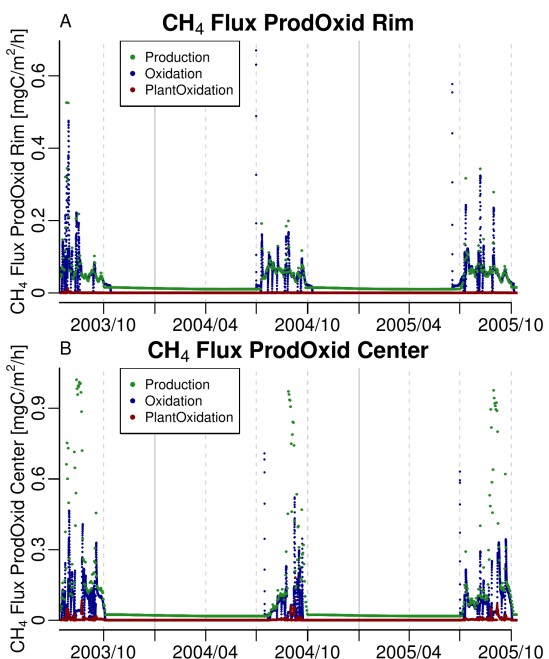

**Figure 5.** Modelled methane amounts that get produced and oxidised at (A) rim and (B) center, split into the different processes. Solid lines indicate 1st of January, dashed lines indicate 1st of April, 1st of July and 1st of October of the respective year. Please note the different scales. The maxima of the values are 0.670 (A) and 1.02 (B) $\mathrm{mgC\,m^{-2}\,h^{-1}}$.

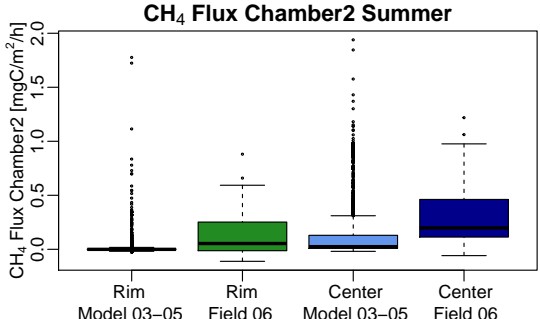

**Figure 6.** Modelled methane flux out of soil at rim and center compared to chamber measurements. Modelled values are only from the summer periods 2003 to 2005, which means less than 5 cm snow are on the ground. Field measurements took place on 39 days from July to September 2006. Because of the widespread of high modelled values, to as high as 86.8 $\mathrm{mgC\,m^{-2}\,h^{-1}}$, a portion of 0.347 % modelled values was cut to provide a reasonable picture. The minimum of the modelled values is -0.0237 $\mathrm{mgC\,m^{-2}\,h^{-1}}$.



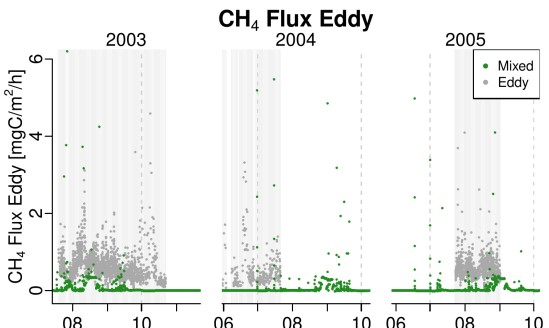

**Figure 7.** Modelled methane flux out of soil in a mixed approach of 65 % rim plus 35 % center compared to eddy covariance measurements. Light grey background indicates measurement data coverage. X-axes indicate 1st day of the respective month of the year. Dashed lines indicate 1st of July and 1st of October of the respective year. Please note the cutouts in-between the different years. Because of the widespread of high modelled values, to as high as 30.4 $mgC\,m^{-2}\,h^{-1}$, a portion of 0.0507 % modelled values was cut to provide a reasonable picture. The minimum of the modelled values is -0.0235 $mgC\,m^{-2}\,h^{-1}$.





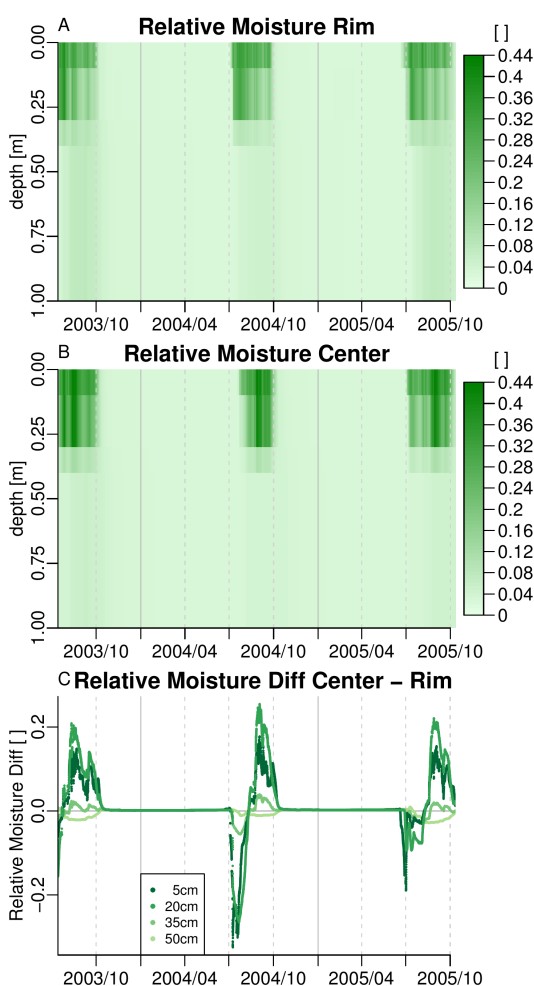

**Figure 8.** Modelled relative soil moisture content of the uppermost metre at (A) rim and (B) center as well as (C) the difference center minus rim in several depths. Solid lines indicate 1st of January, dashed lines indicate 1st of April, 1st of July and 1st of October of the respective year. Scale maximum for (A) and (B) is field capacity, ceiled to two digits.



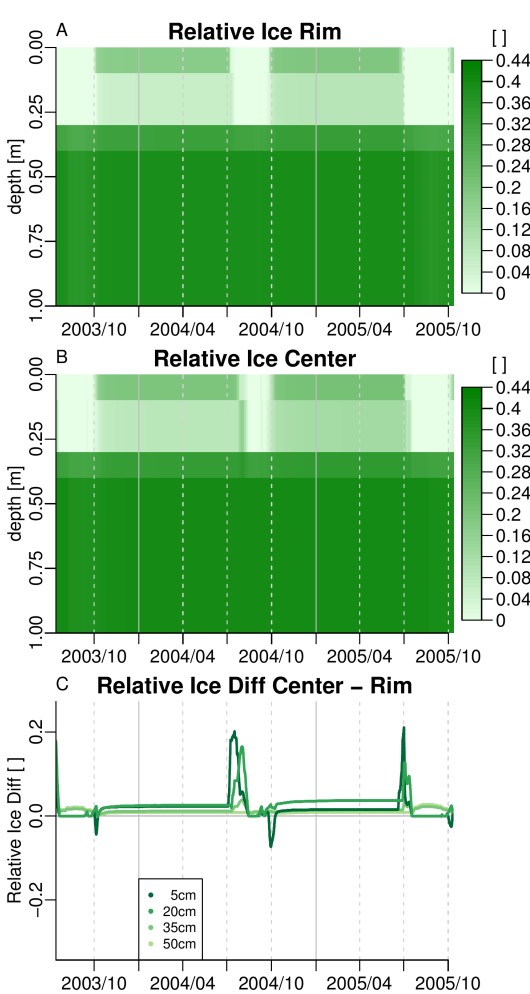

**Figure 9.** Modelled relative soil ice content of the uppermost metre at (A) rim and (B) center as well as (C) the difference center minus rim in several depths. Solid lines indicate 1st of January, dashed lines indicate 1st of April, 1st of July and 1st of October of the respective year. Scale maximum for (A) and (B) is field capacity, ceiled to two digits. The scale for (C) is the same as for the difference of the modelled relative soil moisture content.




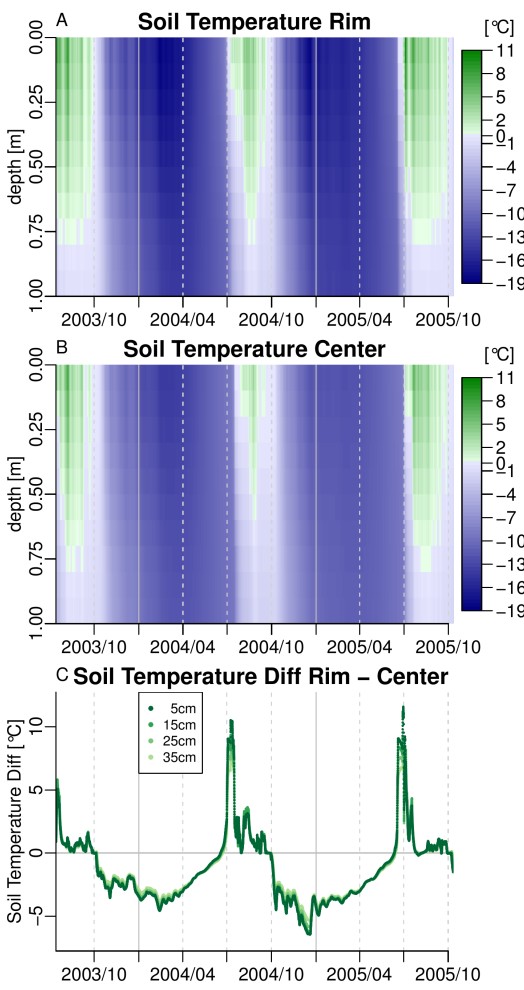

**Figure 10.** Modelled soil temperature of the uppermost metre at (A) rim and (B) center as well as (C) the difference rim minus center in several depths. Solid lines indicate 1st of January, dashed lines indicate 1st of April, 1st of July and 1st of October of the respective year.



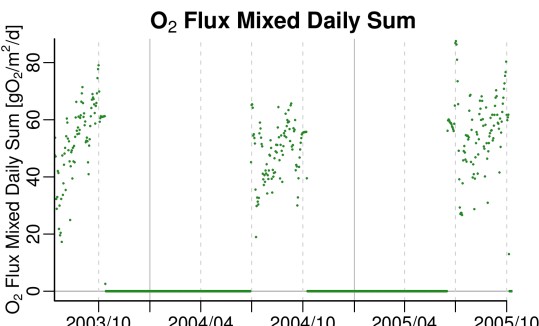

**Figure 11.** Modelled oxygen flux into soil in a mixed approach of 65 % rim plus 35 % center as daily sum. Solid lines indicate 1st of January, dashed lines indicate 1st of April, 1st of July and 1st of October of the respective year. The range of the modelled values is -0.00184 to 87.6 $\mathrm{gO_2\,m^{-2}\,d^{-1}}$.

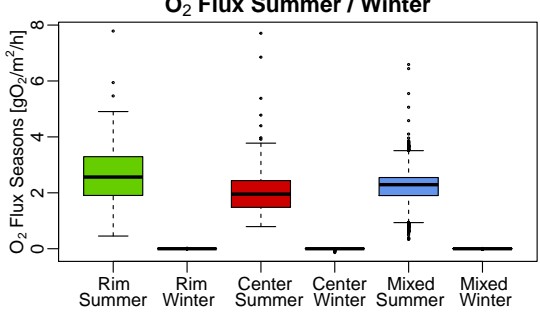

**Figure 12.** Modelled oxygen flux into soil at rim, center and a mixed approach of 65 % rim plus 35 % center, split into summer and winter. Summer means less than 5 cm snow are on the ground, winter is the remainder. Because of the widespread of values, to as high as 16.3 $\mathrm{gO_2\,m^{-2}\,h^{-1}}$, a portion of 0.0118 % values was cut to provide a reasonable picture. The minimum of the values is -0.136 $\mathrm{gO_2\,m^{-2}\,h^{-1}}$.





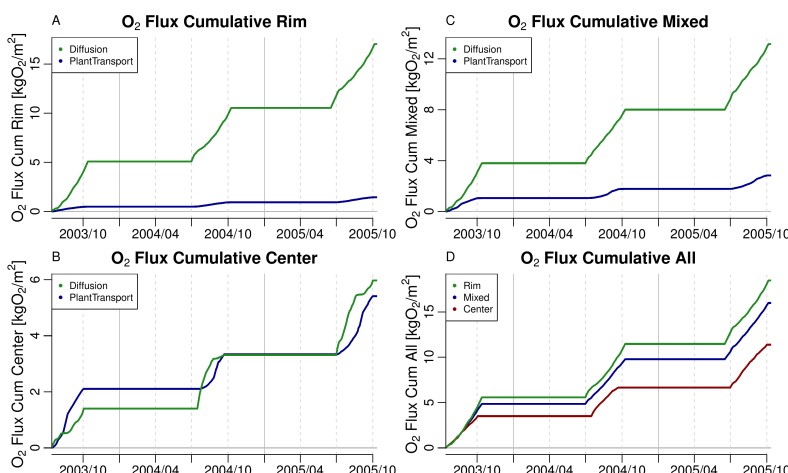

**Figure 13.** Modelled oxygen flux into soil at (A) rim, (B) center, (C) a mixed approach of 65 % rim plus 35 % center, split into the different transport processes, and at (D) rim, center and a mixed approach of 65 % rim plus 35 % center, combined, as cumulative sum. Solid lines indicate 1st of January, dashed lines indicate 1st of April, 1st of July and 1st of October of the respective year. Please note the different scales. Table 4 gives the maximal values.



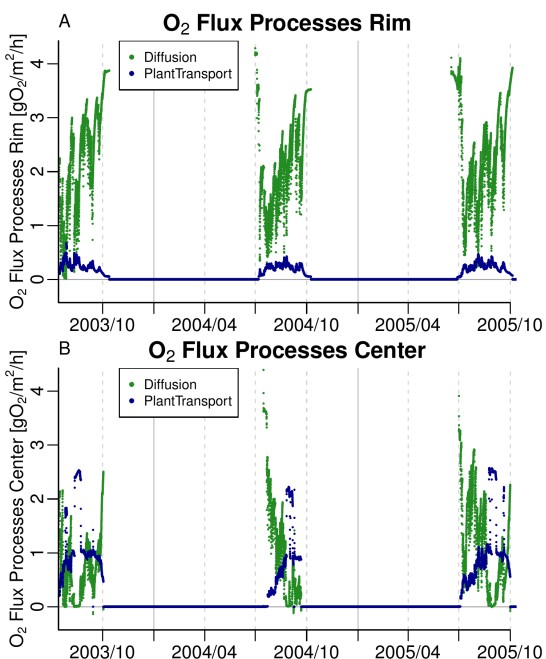

**Figure 14.** Modelled oxygen flux into soil at (A) rim and (B) center, split into the different transport processes. Solid lines indicate 1st of January, dashed lines indicate 1st of April, 1st of July and 1st of October of the respective year. Because of the widespread of high values, to as high as 16.3 (A) and 14.4 (B) $gO_2\,m^{-2}\,h^{-1}$, a portion of 0.0254 % (A) and 0.0178 % (B) values was cut to provide reasonable pictures. The minima of the values are -0.00185 (A) and -0.136 (B) $gO_2\,m^{-2}\,h^{-1}$.