# Peer review of "Process-based modelling of the methane balance in periglacial landscapes (JSBACH-methane)"

_Geoscientific Model Development, 2016_

## Short Comment (SC1) · 6 Jun 2016

Dear authors,

In my role as Executive editor of GMD, I would like to bring to your attention our Editorial version 1.1:

http://www.geosci-model-dev.net/8/3487/2015/gmd-8-3487-2015.html

This highlights some requirements of papers published in GMD, which is also available on the GMD website in the 'Manuscript Types' section:

http://www.geoscientific-model-development.net/submission/manuscript_types.html

In particular, please note that for your paper, the following requirement has not been met in the Discussions paper:

[Figure]

- "The main paper must give the model name and version number (or other unique identifier) in the title."

Please add a version number for JSBACH in the title upon your revised submission to GMD.

Yours,

Astrid Kerkweg
* * *

---

## Referee Comment (RC1) · Anonymous Referee #1 · 8 Jun 2016

The importance of methane balance for Earth System modelling is beyond question. Modelling methane emission and oxidation in the context of Earth System studies is a very complicated task, and accordingly, relatively little progress has been made over the years in this direction. The situation is probably even worse for high latitudes, where very specific processes and phenomena need to be taken into account. Therefore, the work presented here is of importance, and it is clearly a timely effort that must be pursued. In my point of view, it is only a first step towards a really useful methane scheme for periglacial landscapes in JSBACH, but it is a necessary step, and it is a sufficiently large one to warrant publication, provided some revisions detailed below.

- The model is tested only at one single site, for very short periods. There are quite some other usable sites which would have provided a useful, more general picture of the performance and applicability of the model. I would really encourage the authors to

[Figure]

consider using some other sites, such as Zackenberg (e.g. Juncher Jørgensen et al., Nat Geosci 2015, doi:10.1038/NGEO2305) or Barrow (e.g. Zona et al., PNAS 2016, doi:0.1073/pnas.1516017113). In particular, it would be useful to extend the temporal coverage (in terms of seasons, covering shoulder seasons and winter). Late-season processes (see, e.g. Mastepanov et al., Nature 2008, doi:10.1038/nature07464) should also be looked at.

- In the same vein, the paper here talks a lot about springtime emission bursts after snowmelt. Can this actually be seen somewhere?

- I understand that you limited methane exchanges in winter (i.e. when snow > 5 cm) because you had strong methanotrophy. But wouldn't that be naturally limited at very cold temperatures? Why isn't it? As is, your modelled impact of snow is not much more that voluntary error compensation.

- I do not understand why you talk about a 0.5 degree resolution here. Isn't the model setup a site setup? In that case, "spatial resolution" does not make much sense.

- More generally, I would have really liked to see a sensitivity study concerning the parameters of the model, such as in Khvorostiyanov et al. (Tellus, 2008; part 2 of a paper of which you cite part 1. doi:10.1111/j.1600-0889.2007.00336.x). In particular in your case, this would be very useful - you already state that many model parameters are quite uncertain.

- I am not totally convinced that a large-scale evaluation of the model would not make any sense yet, as you state. One could probably gain some understanding of the sensitivity of the model, and possibly see if orders of magnitude of methane fluxes of large scales are OK.

- This scheme is designed by Earth System modelling applications, or at least, it is implemented in a land surface module that was designed for such applications. As for now, you need to run it twice for rims and polygon centers. Clearly, in an ESM context,

this is not a practical solution. I encourage you to discuss ways forward to solve that issue.

- I'm not a native speaker, but to me the use of the English language in the manuscript seems to deserve some improvement.

―――――――――――――――

---

## Referee Comment (RC2) · Anonymous Referee #2 · 12 Jul 2016

General comments

The manuscript extensively documents a new process-based methane module for use with a global land surface scheme. The processes methane production, methane oxidation, methane transport from the soil to the air by ebullition, diffusion and plants are all included in the module. The model has been tested for the Samoylov research site in Siberia, which harbours polygonal tundra consisting of wet centers enclosed by relatively dry rims. Of course the model simulates higher methane emission in the wet centers than in the rims, because of the differences in soil moisture. Methane transport by plants seems to be underestimated, which might be related to uncertainty regarding the parameter values, e.g. root surface properties. Plants may also influence methane emission by root exudates; a source of labile organic carbon brought into the water-saturated layer by sedge roots, from which methane can be produced. I do not know

how important this mechanism can be, but I would like to see some discussion about this role of plants. An interesting finding is the simulated spring burst of methane that has accumulated over winter, although it is not clear if this happens in reality. Overall, a promising methane module, but improvements regarding the hydrological scheme, which currently does not allow for standing water, and fine-tuning of parameter values are required in follow-up studies, before it can be applied on a global scale.

Specific comments

L.1 Remove the word consistent; not clear what is meant and it implies that other methane modules were not consistent.

L.4-6 Is the simulation of oxygen processes new compared to other methane models? If yes, please let the reader know, for example: In this model, oxygen has been explicitly . . .

L.14 Please add a concluding sentence about potential applications of the model.

L.135 It is not clear to me why water content at field capacity is used and not water content at saturation. What soil type is used in these simulations, peat or mineral soil? At least in peaty soils there can be quite some difference between water content at field capacity and saturation. What value has been used for field capacity?

L.398-402 Here I need some more information about soil carbon decomposition and particularly about the depth distribution. Do soil temperature and soil moisture influence the decomposition rate (which would lower the decomposition at depth)? Is there one soil organic carbon pool, or is it divided into labile and more stable carbon pools?

L.393-394 This is confusing: OK, diffusion through the root exodermis is very slow, but this layer is very thin compared to the water layer. Is there still a lower rate of transport through plants than through water?

L.523-524 This was also the conclusion of the previous section. Please combine sections 3.3. and 3.4 as there is quite some overlap.
L.572-574 Isn't this already taken into account by defining saturation as . . . % of field capacity?

Although I am not a native English speaker, the manuscript needs some editing with regard to the English language.

––––––––––––––––––––––––––––––

---

## Author Comment (AC1) · 29 Sep 2016

*Note: in the following document, the original comments made by the reviewer are copied in black, while the authors' responses to these comments follow in blue.*

**Authors' response to comments provided by the Executive editor**

In my role as Executive editor of GMD, I would like to bring to your attention our Editorial version 1.1: http://www.geosci-model-dev.net/8/3487/2015/gmd-8-3487-2015.html

This highlights some requirements of papers published in GMD, which is also available on the GMD website in the 'Manuscript Types' section:
http://www.geoscientific-model-development.net/submission/manuscript_types.html

In particular, please note that for your paper, the following requirement has not been met in the Discussions paper: "The main paper must give the model name and version number (or other unique identifier) in the title.". Please add a version number for JSBACH in the title upon your revised submission to GMD.

The authors thank the Executive Editor for pointing out this missing element in our submitted manuscript. We decided to use 'JSBACH-methane' as our unique model identifier. The title of the revised manuscript will accordingly be modified to "Process-based modelling of the methane balance in periglacial landscapes with JSBACH-methane".

---

## Author Comment (AC2) · 29 Sep 2016

*Note: in the following document, the original comments made by the reviewer are copied in black, while the authors' responses to these comments follow in blue.*

**Authors' response to comments submitted by Reviewer #1**

The importance of methane balance for Earth System modelling is beyond question. Modelling methane emission and oxidation in the context of Earth System studies is a very complicated task, and accordingly, relatively little progress has been made over the years in this direction. The situation is probably even worse for high latitudes, where very specific processes and phenomena need to be taken into account. Therefore, the work presented here is of importance, and it is clearly a timely effort that must be pursued. In my point of view, it is only a first step towards a really useful methane scheme for periglacial landscapes in JSBACH, but it is a necessary step, and it is a sufficiently large one to warrant publication, provided some revisions detailed below.

The authors appreciate the positive evaluation of the reviewer and the support on the publication of this manuscript. We also agree that the presented work represents only a first step towards the establishment of a process-based methane module for JSBACH, and that further improvements will be necessary to upgrade the presented model version for application at larger scales.

The model is tested only at one single site, for very short periods. There are quite some other usable sites which would have provided a useful, more general picture of the performance and applicability of the model. I would really encourage the authors to consider using some other sites, such as Zackenberg (e.g. Juncher Jørgensen et al., Nat Geosci 2015, doi:10.1038/NGEO2305) or Barrow (e.g. Zona et al., PNAS 2016, doi:0.1073/pnas.1516017113). In particular, it would be useful to extend the temporal coverage (in terms of seasons, covering shoulder seasons and winter). Late-season processes (see, e.g. Mastepanov et al., Nature 2008, doi:10.1038/nature07464) should also be looked at.

The authors agree with the reviewer that testing and application of a newly developed model at a single site, using a dataset that exclude large parts of the full annual cycle, can only provide limited insight into the comprehensive applicability of our model. We also agree that both Zackenberg and Barrow would be suitable candidate sites to test the model under different environmental conditions, since both have been studied intensively for a long time, and therefore should be capable to provide the diverse input datasets we require for our model tests. Still, to document the model applicability across sites in the Arctic domain falls out of scope of the presented manuscript. Instead, it is intended here to present the description of the process-based methane module for JSBACH, including a demonstration of model performance using observational datasets from at a single Arctic monitoring site (Samoylov island). Although adding more sites might strengthen the results, it would require investing a large amount of time into observational data collection, quality checks, and additional model runs. Furthermore, it would considerably increase the length of the presented manuscript. The interpretation of model-data-intercomparison would also require many additional paragraphs since it cannot be assumed that we can work with uniform datasets from different sites. Because we also see the need to demonstrate model performance at different Arctic monitoring sites and scales, these steps will be taken for future experiments, and we plan to present them in a follow-up manuscript. Still, the lack of presentation of more sites to evaluate the applicability of the methane model does not undermine the scientific contribution of the presented manuscript. The shortcomings of using just one single site for model performance evaluation will be clearly highlighted in the Discussions section of the revised manuscript. In there, we will also clearly state that further model evaluation at various spatial and temporal scales are required prior to any large-scale application of the model.

- In the same vein, the paper here talks a lot about springtime emission bursts after snow-melt. Can this actually be seen somewhere?

In our response to reviewer #2, we present a list of references where springtime outburst emissions of methane have been documented. Please refer to that letter for details.

- I understand that you limited methane exchanges in winter (i.e. when snow > 5 cm) because you had strong methanotrophy. But wouldn't that be naturally limited at very cold temperatures? Why isn't it? As is, your modelled impact of snow is not much more that voluntary error compensation.

Actually, our simulations yielded cold-season methanotrophy at a very limited level, but continuously lasting throughout the winter. This can be attributed to a suboptimal carbon decomposition routine in the current model implementation. Furthermore, in this model implementation, freezing of relatively dry soils also leads to oxic soil conditions that facilitate methane transport into the soil. Since this process at the found size is not realistic, we used the limited methane exchange as a mechanism to regulate corresponding emissions.

Still, the authors agree that the chosen mechanism that prevents methane release once the snow cover reaches a depth of 5 cm is a very crude approximation of the snow cover influence. The next iteration of the model development will therefore include a more sophisticated, process-based representation of methane diffusion through snow. This upgrade, however, needs to be coupled to a major restructuring of several model components, and therefore cannot be reconciled with the model version presented within the context of this manuscript. We will highlight this shortcoming in the discussion.

- I do not understand why you talk about a 0.5 degree resolution here. Isn't the model setup a site setup? In that case, "spatial resolution" does not make much sense.

Even though we run the model at 'site level' in theory, the general structure of the JSBACH model is still set up for spatially explicit model runs at global scales. In particular, many assumptions behind model structures are exclusively valid at large spatial scales. One prominent example here is the hydrology scheme which works exclusively vertically. This assumption is violated at pedon scale where lateral fluxes from rim to polygon centre dominate soil water content. This is the reason why we split the model experiments into two runs: one for rim and one for centre, in order to mimic this lateral flow. However, there are many more of such assumptions in the model, e.g. for the radiation scheme and energy balance (no south versus north-facing slopes etc.). Since our ultimate target is to provide a new methane module that can be integrated into global scale JSBACH runs, accordingly the structure of our methane module also needs to target spatially explicit runs, and site level runs in fact are landscape-scale spatial runs using input data representing a very small domain. This kind of site-level model evaluation has a long history in land surface modelling, e.g. Sitch et al. (2003); Morales et al. (2005); Beer et al. (2007). We added some new text passages to the revised manuscript to make this clear.

- More generally, I would have really liked to see a sensitivity study concerning the parameters of the model, such as in Khvorostiyanov et al. (Tellus, 2008; part 2 of a paper of which you cite part 1. doi:10.1111/j.1600-0889.2007.00336.x). In particular in your case, this would be very useful - you already state that many model parameters are quite uncertain.

This is a very good point and we completely agree with the reviewer. We reviewed the list of user-defined parameter settings that are required to run the new methane module of JSBACH, and categorized them by relevance and available information to support the chosen settings. Based on this survey, we identified a shortlist of 10 parameters, and defined an uncertainty range of +/- 10 % for their settings. For the resulting maximum and minimum values of this range for each parameter an individual model run was performed, and model sensitivity towards the setting of the chosen parameters was evaluated through changes in the cumulative methane emissions within the study period (Jul 2003 – Oct 2005) that followed the variation of the parameter. Results will be summarized in a new table in the revised manuscript.

- I am not totally convinced that a large-scale evaluation of the model would not make any sense yet, as you state. One could probably gain some understanding of the sensitivity of the model, and possibly see if orders of magnitude of methane fluxes of large scales are OK.

The authors agree with the reviewer that spatially explicit simulations of methane processes could provide new insight into the performance of the new algorithms, since larger grids would cover a range of environmental conditions (climate, land cover, hydrology, freeze-thaw status, etc.). However, this extension again will require a major restructuring of the model algorithms, particularly related to conflicts of the new methane algorithms with the existing hydrology implementation in JSBACH. This goes beyond the scope of the presented manuscript. A larger scale evaluation of the model, based on spatially-explicit runs in a regional setting, is therefore planned for a follow-up study, see also the discussion above.

- This scheme is designed by Earth System modelling applications, or at least, it is implemented in a land surface module that was designed for such applications. As for now, you need to run it twice for rims and polygon centers. Clearly, in an ESM context, this is not a practical solution. I encourage you to discuss ways forward to solve that issue.

The authors agree with the reviewer that separate model runs for polygon rims and polygon centers can only be a work-around when the target is to evaluate a newly developed methane module for JSBACH that ultimately can be applied in global simulations. Still, we believe that this solution is very effective for the purpose of the presented manuscript, i.e. to demonstrate the performance of the algorithm under strongly different soil moisture regimes. For future applications, we are already in the process of adopting the methane module to large-scale models of water-logged conditions (TOPMODEL approach). These upgraded model versions will be presented in a follow-up manuscript.

- I'm not a native speaker, but to me the use of the English language in the manuscript seems to deserve some improvement.

We plan to have the revised version of the manuscript reviewed by a native speaker before resubmission.

**References**
Beer, C., Lucht, W., Gerten, D., Thonicke, K., and Schmullius, C.: Effects of soil freezing and thawing on vegetation carbon density in Siberia: A modeling analysis with the Lund-Potsdam-Jena Dynamic Global Vegetation Model (LPJ-DGVM), Glob. Biogeochem. Cy., 21, GB1012, 2007.

Morales, P., Sykes, M. T., Prentice, I. C., Smith, P., Smith, B., Bugmann, H., Zierl, B., Friedlingstein, P., Viovy, N., Sabaté, S., Sánchez, A., Pla, E., Gracia, C. A., Sitch, S., Arneth, A., and Ogee, J.: Comparing and evaluating process-based ecosystem model predictions of carbon and water fluxes in major European forest biomes, Glob. Change Biol., 11, 2211-2233, 2005.

Sitch, S., Smith, B., Prentice, I. C., Arneth, A., Bondeau, A., Cramer, W., Kaplan, J. O., Levis, S., Lucht, W., Sykes, M. T., Thonicke, K., and Venevsky, S.: Evaluation of ecosystem dynamics, plant geography and terrestrial carbon cycling in the LPJ dynamic global vegetation model, Glob. Change Biol., 9, 161-185, 2003.

---

## Author Comment (AC3) · 29 Sep 2016

*Note: in the following document, the original comments made by the reviewer are copied in black, while the authors' responses to these comments follow in blue.*

**Authors' response to comments submitted by Reviewer #2**

General comments

The manuscript extensively documents a new process-based methane module for use with a global land surface scheme. The processes methane production, methane oxidation, methane transport from the soil to the air by ebullition, diffusion and plants are all included in the module. The model has been tested for the Samoylov research site in Siberia, which harbours polygonal tundra consisting of wet centers enclosed by relatively dry rims. Of course the model simulates higher methane emission in the wet centers than in the rims, because of the differences in soil moisture. Methane transport by plants seems to be underestimated, which might be related to uncertainty regarding the parameter values, e.g. root surface properties. Plants may also influence methane emission by root exudates; a source of labile organic carbon brought into the water-saturated layer by sedge roots, from which methane can be produced. I do not know how important this mechanism can be, but I would like to see some discussion about this role of plants.

The reviewer certainly raises a very important point regarding the influence of vegetation on the methane emissions. The reviewer is right that the $CH_4$ transported by plants is limited by the parameter values related to the plant characteristics. As requested by reviewer #1, in the revised paper we are incorporating results of a suite of sensitivity experiments, which also include the variation of the density of vascular plants. The effect of the range of parameter values is evaluated and discussed in the context of the total methane emissions.

With regard to the methane production and emission due to root exudates, we thank the reviewer for commenting on this process. The contribution of root exudates to methane production and emission has been highly neglected in models, and is also not considered in our model. This process has been also understudied in field experiments because it is a process that cannot be measured directly.

The indirect contribution to methane production from root exudates has been mainly analyzed in rice fields (Neue et al., 1996) and to less extent in peat soils (Koelbener et al., 2010). The later study suggests that the rate of root exudate is linked to the nutrient availability in soils, in a way that more root exudates are present in plants located in nutrient poor wetland soils. The wetland soils in Arctic tundra are known to be nitrogen limited (Melle et al., 2015; Gurevitch et al., 2002). In a recent work by Beermann et al. (2015), the authors evaluate the nitrogen content in different compartments of soil and vegetation in a polygonal lowland tundra located in Indigirka, Russia; the authors findings revealed that plant growth in these environments are co-limited by nitrogen and phosphorous with only about 5 % of the total nitrogen soil content to be an active part of the biological fraction. Knoblauch et al. (2015) investigated the indirect role of root exudates for methane production in polygonal ponds and water-saturated soils in Samoylov Island. The authors found almost 4 fold higher potential methane production rates in vegetated sites than in non-vegetated sites, both with same C and N concentrations in the soil, indicating the potential important contribution of root exudates to methane soil production. Perhaps the most direct attempt to quantify this process has been made by Ström et al. (2012), who studied the impact of plant exudates on methane emissions in an Arctic wetland in Greenland, and related them to different vascular plants species. They concluded that the presence of vascular plants in Arctic wetlands support the production of low molecular weight carbon compounds that are highly labile, which can promote methane emissions through their methanogenic decomposition.

Thus, with no doubt the contribution of root exudates to methane emissions can play an important role in Lena Delta, and needs to be taken into account in models. Of particular importance is to consider potential nutrient mobilization in Arctic soils related to permafrost

degradation under climate change (Kuhry et al., 2010). This effect may enrich wetland nutrient contents from mesotrophic to eutrophic state, implying a minor contribution to methane production due to root exudates.

Unfortunately, in the current configuration of the JSBACH model, the concentration of nutrients in the soil is not taken into account, thus the nutrients assimilation by plants roots is also not considered. These processes need to be parameterized in the model before the root exudate contribution to the total methane emissions can be accounted. We expect that this process is included in the future model iterations, and in that way the methane module can be completed with the inclusion of root exudates. We will discuss this point in the same way as above in the revised version of this manuscript.

An interesting finding is the simulated spring burst of methane that has accumulated over winter, although it is not clear if this happens in reality.

The authors agree with the reviewer that more tests on this effect need to be performed to properly evaluate how realistic are the spring burst as seen in the model output. Several published studies have reported the presence of methane spring thaw burst in wetlands and lakes at high latitudes using different measurement techniques (e.g. eddy covariance, chambers). These events are sporadic, and few studies have been exclusively dedicated to investigate spring methane emissions in the transition from winter to the growing season (e.g. eddy covariance technique in a lake and wetland area in north Sweden, Jammet et al., 2015; using chamber measurements and eddy techniques in north Sweden, Friborg et al., 1997; or in Finland, Hargreaves et al., 2001; chamber measurements in a peatland in northern Japan, Tokida et al., 2007; chamber measurements in a wetland area in Northeast China, Song et al., 2012). These few studies suggest that the methane spring thaw emissions occur over short periods, and the magnitude of the short-term emissions can exceed the mean summer fluxes by a factor of 2 to 3. These emission events may therefore account for a large share of the total annual emissions. However, both the duration and magnitude of the spring thaw events are to date highly uncertain. Not only a further evaluation in the model is needed to evaluate the representativeness of these events in the model output, but also more observational data measured during the spring thaw season are needed to adequately characterize these events. For the model side, these steps will be considered in a follow-up manuscript in which we will also apply the methane module to a larger spatial scale. We will include this discussion in the revised version of the manuscript.

Overall, a promising methane module, but improvements regarding the hydrological scheme, which currently does not allow for standing water, and fine-tuning of parameter values are required in follow-up studies, before it can be applied on a global scale.

The authors appreciate the support of the reviewer to our manuscript. As is correctly pointed out, the status of the configuration presented in the current manuscript still requires improvements toward the hydrological scheme. This will be particularly important for any large-scale application in future. Practically, we see a possibility to apply a TOPMODEL approach for estimating the inundated fraction of the grid cell. These steps are currently been taken into account and will be part of a follow up study. Please see also our discussion with reviewer #1 in this respect. In addition, before the methane module can be applied on a global scale, current tests at a regional scale will be implemented. However, we would like to stress that the intention of the current manuscript is the presentation and description of the newly developed process-based methane module, and its application at a site level scale. This should be always the first step before going global. Therefore including regional applications in this work goes beyond the scope of this manuscript.

Specific comments

L.1 Remove the word consistent; not clear what is meant and it implies that other methane modules were not consistent.

In this context, what we wanted to say was that the new model is even more process-based than previously existing model implementations, e.g. regarding the use of physical state variables (water or ice content, free pore space) in modules such as the computation of thermal conduction, or gas transport. We will avoid a misleading use of the word 'consistent', as suggested by the reviewer. The manuscript text will be revised accordingly.

L.4-6 Is the simulation of oxygen processes new compared to other methane models?
If yes, please let the reader know, for example: In this model, oxygen has been explicitly

Yes, one of the novel developments of the methane model presented in this manuscript is the explicit addition of oxygen content in the soil as a state variable, and of two methane oxidation processes. As discussed in the manuscript, the two oxidation processes are:

1) bulk soil oxidation of methane: here, above the water table at most 40 % of the available oxygen is used for heterotrophic respiration, and 10 % is reduced by other physical processes such as N or P oxidation. Below the water table, 50 % of the oxygen content is available for methane oxidation. Via Michaelis-Menten kinetics for methane and oxygen as well as methane and oxygen availability, the model then calculated how much methane is oxidized;

2) oxidation of the methane that is available to be transported by roots (plant oxidation): the model allows oxygen also to enter through the aerenchyma of the plants, and this oxygen is then allowed to oxidize the available methane to be transported by roots. Again, the same procedure like for bulk soil oxidation calculates how much methane is actually oxidized.

In the revised manuscript, we will add explicit sentences on the addition of the oxygen processes to emphasize the novelty of this configuration. This explicit consideration of oxygen is an important point for making the model more process-based where previously these processes have been implicitly taken into account by effective parameters.

L.14 Please add a concluding sentence about potential applications of the model.

As suggested by the reviewer, we will add in the revised manuscript a concluding sentence about the potential applications for the methane model. To mention some of them: application in studies at regional to global scale with the main focus of evaluating the contribution of methane emissions to the global carbon budget due to permafrost thaw in high latitudes, the projection of future methane emissions (total and from the different emission processes) at different spatial scales (from regional to global), and ultimately a quantification of future methane emission contributions on future climate and corresponding feedback mechanisms.

L.135 It is not clear to me why water content at field capacity is used and not water content at saturation. What soil type is used in these simulations, peat or mineral soil?
At least in peaty soils there can be quite some difference between water content at field capacity and saturation. What value has been used for field capacity?

This is a good point and from a site-level perspective this point is also very relevant. Looking at methane emissions from polygon rim or centre, pedon-scale hydrology plays an important role, and in particular lateral water fluxes. As the ultimate goal is to arrive at a methane module embedded into a global land surface scheme, from the beginning this methane module has been incorporated into the existing land surface model, which uses a simplified hydrological scheme, e.g. assuming lateral flows can be neglected at 0.5 degree grid cell size. Still, we believe that running the model using site-level climate data and soil information is very useful for a first-order guess about its reliability. Therefore, we mimic as much as possible the different hydrological regimes at rim and centre with a different simulation protocol, and also we apply a semi-empirical relationship of relative soil moisture to water table depth. The reason is that the hydrology scheme of the land-surface model does not work with saturation of soils but water above field capacity is basically removed by runoff. This representation of

water table height will be obsolete in future when coupled to hydrology schemes such as e.g. TOPMODEL, which will estimate the inundated fraction of each grid cell. We will revise the manuscript in order to clarify this point.

Regarding the question on soil type, the current model implementation just considers mineral soil, no peat layers in this version. Field capacity was set to 0.435, porosity to 0.448.

L.398-402 Here I need some more information about soil carbon decomposition and particularly about the depth distribution. Do soil temperature and soil moisture influence the decomposition rate (which would lower the decomposition at depth)? Is there one soil organic carbon pool, or is it divided into labile and more stable carbon pools?

The processes the reviewer refers to are explicitly resolved along the vertical soil profile, with detailed information on the implementation provided in the appendices of the submitted manuscript. The carbon decomposition is temperature dependent, but moisture dependency has been implemented only indirectly: The soil profile is divided into three sections, and each of them has its own temperature used for the decomposition. The vertical extent of each section depends on the positions of actual and the minimal water table, respectively, which is where the moisture influence comes in.

Details on the depth distribution of the existing soil carbon are provided in Appendix 3 of the manuscript. Because for the carbon decomposition the soil profile is only split up into three sections, the decomposition module itself does not use this information at the original, high resolution. But the calculated decomposed carbon amounts of the three pools are then distributed by the methane module over the layers according to the distribution of the soil carbon as described in Appendix 3. Further details on these processes are also shown in Appendix 2.

L.393-394 This is confusing: OK, diffusion through the root exodermis is very slow, but this layer is very thin compared to the water layer. Is there still a lower rate of transport through plants than through water?

In the methane module, the diffusion of gas through root exodermis is represented as only 80 % of the diffusion of gas in water. Although the exodermis is a very thin layer, it is an efficient barrier against gas exchange that allows the gases to remain within the plant (e.g. in case of oxygen) to use it for metabolic processes. Thus, the diffusion of methane from the soil pores through this very thin layer cannot be as fast as in water. This is explained in section 2.2.7, L.311-330 of the discussion manuscript.

L.523-524 This was also the conclusion of the previous section. Please combine sections 3.3. and 3.4 as there is quite some overlap.

These sections will be combined in the revised manuscript as suggested by the reviewer.

L.572-574 Isn't this already taken into account by defining saturation as … % of field capacity?

The 90 % field capacity in the soil corresponds only to the soil layer, and in this current model configuration no standing water above the soil surface is taken into account. Despite the soil remaining moist, there will be no methane emissions to the atmosphere. By considering this in the context of comparing chamber measurements to model results, we wrote the statement that the model might have "less moist hydrological conditions" than the field sites where the chamber measurements were performed. In the chamber measurements sites, potential accumulation of water above the soil can lead to higher methane emissions, thus contributing to a larger discrepancy when comparing field measurements to model results. We will clarify this sentence in the revised version of the manuscript.

Although I am not a native English speaker, the manuscript needs some editing with regard to the English language.

As suggested also to reviewer #1, we plan to have the revised version of the manuscript reviewed by a native speaker before resubmission.

**References cited in this response.**

Breermann, F., A. Teltewskoi, C. Fiencke, E-M, Pfeiffer and L. Kutzbach, 2015. Stoichiometric analysis of nutrient availability (N, P, K) within soils of polygonal tundra. *Biogeochemistry*, 122: 211-227, doi: 10.1007/s10533-014-0037-4.

Friborg, T., T.R. Christensen and H. Sogaard, 1997. Rapid response of greenhouse gas emission to early spring thaw in a subarctic mire as shown by micrometeorological thechniques. *Geophysical Research Letters,* 24:23, 3061-3064.

Gurevitch, J., S. Scheiner and G. Fox, 2002. The Ecology of Plants, Sinauer Associates, Inc., Sunderland, 2nd edition.

Hargreaves, K.J., D. Fowler, C.E.R. Pitcairn and M. Aurela, 2001. Annual methane emission from Finnish mires estimated from eddy covariance campaign measurements. *Theor. Appl. Climatol.*, 70: 203-2013.

Jammet, M., P. Crill, S. Dengel and T. Friborg, 2015. Large methane emissions from a subarctic lake during spring thaw: mechanisms and landscape significance. *Journal of Geophysical Research: Biogeosciences*, 120, 2289-2305, doi: 10.002/2015JG003137.

Knoblauch, C., O. Spott, S. Evgrafova, L. Kutzbach and E-M, Pfeiffer, 2015. Regulation of methane production, oxidation, and emission by vascular plants and bryophytes in ponds of the northeast Siberian polygonal tundra. *Journal of Geophysical Research: Biogeosciences*. 120: 2525-2541, doi: 10.1002/2015JG003053.

Koelbener, A., L. Ström, P.J. Edwards and H.O. Venterink, 2010, Plant species from mesotrophic wetlands cause relatively high methane emissions from peat soils. *Plant Soil*, 326:147-158, doi: 10.1007/s11104-009-9989-x

Kuhry, P., E. Dorrepaal, G. Hugelius, E.A.G. Schuur, and C. Tarnocai, 2010. Potential remobilization of belowground permafrost carbon under future global warming. *Permafrost and Periglacial Processes*, 21(2):208–214.

Melle, C., M. Wallenstein, A. Darrouzet-Nardi and M.N. Weintraub, 2015. Microbial activity is not always limited by nitrogen in Arctic tundra soils. *Soil Biology and Biochemistry,* 90:52-61, doi: 10.1016/j.soilbio.2015.07.023.

Neue, H.U., R. Wassmann, H.K. Kludze, W. Bujun and R.S. Lantin, 1997. Factors and processes controlling methane emissions from rice fields. *Nutrient Cycling in Agroecosystems,* 49: 111-117.

Song, C., X. Xiaofeng, X. Sun, H. Tian, L. Sun, Y. Miao, X. Wang and Y. Guo, 2012. Large methane emission upon spring thaw from natural wetlands in the northern permafrost region. *Environmental Research Letters,* 7, 8 pp, doi: 10.1088/1748-9326/7/3/034009.

Ström L., T. Tagesson, M. Mastepanov and t. R. Christensen, 2012. Presence of *Eriophorum schuchzeri* enhances substrate availability and methane emission in an Arctic wetland. *Soil Biology and Biochemistry*, 45:61-70, doi: 10.1016/j.soilbio.2011.09.005.

Tokida, T., M. Mizoguchi, T. Miyazaki, A. Kagemoto, O. Nagata and R. Hatano, 2007. Episodic release of methane bubbles from peatland during spring thaw. *Chemosphere,* 70:165-171, doi: 10.1016/j.chemosphere.2007.06.042.

---

## Author Response (AR1)

MPI for Biogeochemistry • P.O.Box 10 01 64 • 07701 Jena

Sonja Kaiser

phone  +49 (0)3641 57 6216
Fax     +49 (0)3641 57-7300

E-Mail: skaiser@bgc-jena.mpg.de
Web    www.bgc-jena.mpg.de

Directors
Prof. Susan E. Trumbore, PhD (Manag. Dir.)
Prof. Dr. Martin Heimann
Prof. Dr. Markus Reichstein

2016-10-27

**Submission of a revised manuscript (GMD-2016-103)**

Dear Dr. Arndt,

the submitted text versions of the manuscript GMD-2016-103 contain all changes that we previously announced to work in, based on the reviewers' comments. As requested, together with the 'clean' text version we submitted a 'track changes' version where edited sections are highlighted in red font. The revision has been discussed intensively among the co-authors, therefore the revised text version represents the opinion of the entire author team.

We discussed the language of the manuscript within the past weeks with members of the Copernicus editorial office, namely Svenja Lange and Meredith Gunnells. They confirmed that they consider the current language of the text as high quality, and that remaining flaws in the language can be taken care of by the standard 'light copy editing' that is offered by Copernicus for all accepted manuscripts. We therefore decided not to conduct a professional language check of our own. Since we announced to do just that in our answer letters to the reviewers, we would appreciate if you could communicate this issue to them, in case they request to review the revised manuscript version again.

Moreover, in this current version of the manuscript, we decided to change the order of authors slightly. Could you please also change the author list in your processing files accordingly?

Yours faithfully, Sonja Kaiser

Hans-Knöll-Straße 10          Tel.: +49 (0)3641 57-60          ID-Nr. DE 129517720
07745 Jena                    Fax:  +49 (0)3641 57-70          www.bgc-jena.mpg.de
Germany

[Figure]

Manuscript prepared for Geosci. Model Dev.
with version 2015/04/24 7.83 Copernicus papers of the LaTeX class copernicus.cls.
Date: 27 October 2016

**Process-based modelling of the methane balance in periglacial landscapes  (JSBACH-methane)**

S. Kaiser[1], M. Göckede[1], K. Castro-Morales[1], C. Knoblauch[2], A. Ekici[1,3],
T. Kleinen[4], S. Zubrzycki[2], T. Sachs[5], C. Wille[2], and C. Beer[6]

[1]Max Planck Institute for Biogeochemistry, Jena, Germany
[2]Universität Hamburg, Hamburg, Germany
[3]Uni Research Climate, Bjerknes Centre for Climate Research, Bergen, Norway
[4]Max Planck Institute for Meteorology, Hamburg, Germany
[5]Helmholtz Centre Potsdam GFZ German Research Centre for Geosciences, Potsdam, Germany
[6]Stockholm University, Stockholm, Sweden

*Correspondence to:* S. Kaiser (skaiser@bgc-jena.mpg.de)

[revised manuscript text omitted]

$$c_{plox}^{CH_4} = \min\left( V_{max} \cdot \frac{f_r \cdot c^{CH_4}}{K_m^{CH_4} + f_r \cdot c^{CH_4}} \cdot \frac{c_{plant}^{O_2}}{K_m^{O_2} + c_{plant}^{O_2}} \cdot Q_{10}^{\frac{T-10}{10}} \cdot \mathrm{d}t,\ 2 \cdot c_{plant}^{O_2},\ f_r \cdot c^{CH_4} \right) . \tag{8}$$

The variables' definitions are the same as for the bulk soil methane oxidation, $f_r$ is the fraction of roots in the layer that are able to transport gases, and $c_{plant}^{O_2}$ is the concentration of oxygen transported by plants. Carbon and oxygen pools are adjusted accordingly. The total exchange with the atmosphere is determined by summing the total amount of gas that is calculated by multiplying the concentrations by their pore space.

**2.3 Simulation setup**

As a global land surface scheme,  the JSBACH model is set up for spatially explicit model runs at larger scales. Accordingly, many assumptions behind the model structures are only valid at large spatial scales. One prominent example here is the hydrology scheme, which works exclusively vertically, therefore cannot represent lateral water flow  from rim to center, which is a process of major importance in polygonal tundra sites.  Other examples include assumptions regarding e.g. the modules for radiation scheme and energy balance (no south- versus north-facing slopes etc.). Since our ultimate target is to provide a new methane module that can be integrated into global scale JSBACH simulations, accordingly the structure of our methane module also needs to target spatially explicit experiments. Thus, the site level runs presented here are landscape-scale spatial runs with a grid cell size of 0.5 ° using input data representing a very small domain.

To still facilitate site-level simulations that capture the general hydrologic characteristics of a polygonal tundra site, we split the model experiments into two separate runs, one for rim and one for center. A redistribution of excess water from the rim area into polygon centers was added in order to mimic lateral flow. In more detail, the performed experiment consisted of two  simulation runs with

different settings for rim and center. The polygon rim is assumed to be a normal upland soil, and a standard JSBACH simulation run was performed. For the polygon center, runoff and drainage of the rim have been collected and added to center precipitation. Additionally, for the center run, runoff and drainage have been switched off until the soil water content reached field capacity.

The sequence of methane processes executed in the module is identical to the above described order within Sect. 2.2.1 to 2.2.8, and has been sorted according to the velocity of the specific processes, from fast to slow. The impact of changing this sequence on total and component methane flux rates was tested in a separate sensitivity study (not shown). These tests indicated only a minor influence of the sequence to the partitioning of the fluxes between the transport processes compared to the influence of hydrology or the definition of the processes themselves. Still, it cannot be  ruled out that modelled methane processes may be  modified through the chosen order under certain conditions.

The carbon pools for rim and center were initialised using data from Zubrzycki et al. (2013) and information from Harden et al. (2012), Schirrmeister et al. (2011) and Sachs et al. (2010). The used values for rim and center for Samoylov are 627.61 $\mathrm{mol\,m^{-2}}$ and 731.94 $\mathrm{mol\,m^{-2}}$ for the upper carbon pool (i.e. the two zones making up the unsaturated and temporarily saturated soil layers) and 16355 $\mathrm{mol\,m^{-2}}$ and 25424 $\mathrm{mol\,m^{-2}}$ for the lower carbon pool (i.e. the permanently saturated zone). Because of the lack of information on how the modelled soil carbon from these two pools is distributed vertically, a depth distribution is applied to the decomposed carbon instead. For all layers within one stratum, equal decomposition velocity is assumed. The relative amounts of measured carbon are applied as distribution aid for the decomposed carbon. The layers used were 10 cm in height.  The only further settings varying between rim and center are two vegetation parameters required for the process of plant transport, i.e. the number of tillers per square metre, and the dominance of *Carex aquatilis*. Beyond the definitions cited above, the model has not been calibrated to site specific processes or properties.

To  initialise hydrological conditions, a spin-up  of 100 years was done using one single year of climate data with average conditions from the period of observations. Starting in year 41 of this spin-up, the methane processes  were activated. This setup was chosen to stabilize the hydrological conditions before the methane processes were  included. After finalising the spin-up, the time period of interest has been calculated with actual climate data.

**2.4 Sensitivity experiments**

We reviewed the list of parameters that are required to run the new methane module of JSBACH and categorised them by relevance and available information to support the chosen settings. Based on this survey, we identified a shortlist of 10 parameters, which are listed in Table 2. To allow for a uniform processing of all parameters on this list, we assumed an uncertainty range of +/- 10 % for each of these settings. Changing each parameter by these percentages and performing for each of those an individual model run yielded a range of resulting methane emissions according to the influence of each parameter. Model sensitivity towards the setting of the chosen parameters was evaluated through changes in the cumulative methane emissions over the modelled time period that followed the variation of the parameter.

**2.5 Forcing and evaluation data**

[revised manuscript text omitted]

**3.3  Role of different transport processes**

During most  part of the year, the diffusive methane flux is rather small at the rim (Fig. 3A) and sometimes slightly negative at the center (Fig. 3B).  The largest methane emissions, both at the rim and at the center,

occur during spring. In this season, the methane that  is produced in the top soil from late autumn on and accumulated during winter is released in the form of so-called spring bursts upon snow thaw.

515

Plant mediated methane transport is smaller than diffusion but more pronounced at the center than at the rim (Fig. 3A and 3B) because plant transport was defined to be slower than diffusion in water and it should thus lead to lower emissions under less wet conditions.  Despite the exodermis is a very thin layer, it is an efficient barrier against

520 gas exchange, maintaining gases such as oxygen that are necessary for metabolic processes inside the roots. Thus, the diffusion rate through roots is slower than through water, and in turn, diffusion in water is much slower than diffusion in air. Moreover, the soils in the center were not water saturated in the model, promoting diffusive methane released though coarse pores. Under wet soil conditions, plant transport is dominant relative to diffusion

525  , because diffusion in water is a slower process. At the center, ebullition is the most important process  (Fig. 3B)  while diffusion at the rim (Fig. 3A). This is due to the drier conditions at the rim that allow a fast diffusion through air, while ebullition is only possible  under conditions of high water content. Because in the model, higher soil moisture is calculated from the middle to the end of the thawing season, most

530 of the emissions by ebullition and plant transport occur at the center  (Fig. 3B).

In the mixed approach, only the diffusion of the rim alters substantially the pattern of the emissions  (Fig. 3C). In total, the polygon center accounts for a 6.8 times as large fraction of emissions as the rim due to the higher methane production under wetter conditions (Fig. 3D). This

535 means, a total share of 78.6 % of the methane emissions in the mixed approach is coming from the center. Emissions at the rim are highest during spring, while they are highest at the center during the mid and late season (Fig. 3D).

When comparing the total fluxes of the center to the ones of the rim, diffusion is almost doubled,

540 plant transport is 19 times as high, and ebullition is 18 times as high (Table 1). This results in almost seven times higher total methane emissions at the center than at the rim.  At the rim diffusion is more than 13 times as high as plant transport  , while at the center it is just slightly higher than   plant transport. Ebullition is about 4.5 times as high as plant transport both at the rim and at the center. These differences are again due to

545 the differences in soil moisture content, which allow more production under higher soil moisture and  leads to more methane emissions.  Thus, under drier conditions, diffusion in air will transport the main portion  of gas, and

under wetter conditions  plant transport may increase relative to diffusion. With reduced soil air, the remaining velocity of the diffusion is almost  the same order of magnitude  than the overall velocity of plant transport, in contrast to the velocity of diffusion mainly through air.

~~Still it seems, that the plant transport in the model is too low compared to the total flux. While the diffusion flux to the atmosphere only happens at the soil surface, the surface area of the gas transporting roots is the relevant boundary for plant transport. The value of this is not well-known, so the module might need further adjusting of parameters connected to plant root surface area to improve the share of plant transport. Furthermore, ebullition needs substantial amounts of soil moisture, and this is more common at the center than at the rim. Consequently, substantially more ebullition is found at the center than at the rim. In the mixed approach, diffusion accounts for about 2.5 times of the emissions of plant transport, while ebullition accounts for 4.5 times of it. Overall, 0.588 of carbon are emitted by each square metre during the modelled time period from 14 July 2003 to 11 October 2005.~~

**3.4**

 Not only splitting the total methane flux into several transport processes  allows evaluating the relative contribution of each process linked to rim or center characteristics, but also  it is possible to analyse differences in temporal patterns (Fig. 4A).  As noted above, at the rim the fluxes are much lower  than at the center (Fig. 4B) because  less methane is produced  under drier conditions, or methane becomes oxidised in the soil column. Ebullition makes up a large portion of the total budget at both microsites at isolated time steps, reflecting the nature of this process, while its total amount for rim is rather small over longer timeframes. At the rim, diffusion represents both the second largest methane release and substantial uptake during the season (Fig. 4A). The smallest flux portion at the rim is due to plant transport, which also shows some uptake. In contrast, at the center plant transport plays a much more pronounced role, and diffusion fluxes are more negative. All these effects occur in the different hydrological regimes at the rim and at the center.seems, that the plant transport is too low~~

Furthermore, ebullition can only take place in soils with high soil moisture content, and this is more common at the center than at the rim. Consequently, substantially more ebullition is found at the

585 center than at the rim. In the mixed approach, diffusion accounts for about 2.5 times of the emissions of plant transport, while ebullition accounts for 4.5 times of it. Overall, 0.588 g of carbon are emitted by each square metre during the modelled time period from 14 July 2003 to 11 October 2005.

**3.4  Parameter sensitivity tests**

590 Results of the sensitivity tests are summarised in Table 2 and indicate that just one of the chosen parameters, fracCH4Anox, has a major influence on the cumulative methane emissions when varied within a 10 % range. FracCh4Anox represents the fraction of methane produced under anaerobic conditions compared to the total decomposition flux.

595 For two more parameters, fracO2forOx+fracO2forPh and KmO2, the net effect was still larger than 1 percent. FracO2forOx+fracO2forPh influences the available amount of oxygen for the methane oxidation, whereas KmO2 influences the oxidation as Michaelis–Menten constant for oxygen. For all remaining parameters, only negligible effects on the cumulative methane emissions were found.

**3.5  Production versus oxidation**

[revised manuscript text omitted]

645 For this comparison, the same constraints hold like for the comparison to chamber data. The modelled fluxes  differ from field measurements because of  differences in thermal or hydrological conditions.  Critical are periods where observations show substantial methane emissions while at the same time  model results only show minor emissions, e.g. in autumn
650 2003 or spring 2004. During these periods, modelled soil temperature values below zero and snow cover result in modelled methane fluxes of virtually zero, while in reality soils might be warmer and gas diffusion through snow might

655   be possible (see discussion section).

Still, Fig. 7 also shows some patterns that are present in both model results and observations, e.g. periods with increasing fluxes that are followed by a sudden decline in the fluxes in a cyclic manner during a single season.  These patterns are linked to the changing soil moisture content. Unfor-
660   tunately only the first season is covered well by field measurements, while the second misses the later part, and the third covers just a part within.  The model shows the largest methane emissions during spring upon snow thaw for both rim and center in the form of burst. There is still little evidence in
665   field measurements of the occurrence and magnitude of spring bursts, and to our knowledge no published data on this effect exists for Samoylov Island. In the discussion section, we briefly review the evidence of spring bursts in other northern wetland areas to evaluate the representativeness of these events in the model results.

670   For additional results concerning modelled oxygen uptake, such as mixed daily sum, seasonally split and cumulative sums as well as transport process split, see App. B4.

**4 Discussion**

This paper aims to present the  structure of a newly developed methane module for the land surface scheme JSBACH
675    and evaluate its general performance against field observations. The new module itself is completely integrated into the  larger framework of the JSBACH model, therefore sensitivity tests can only be conducted using the full model and a clean separation between existing structure and new components is not always
680   possible. The interpretation and discussion of all findings should therefore consider  that the functioning of the new methane module is to a large part dependent on, and in many aspects limited by, the performance of the JSBACH model  as a whole.

685   The presented methane module determines production, oxidation and transport of methane to the at- mosphere.  All of these key processes are heavily dependent on soil water status as well as the quality and quantity of carbon in different soil pools. Both of these aspects, i.e. soil hydrology and carbon decomposition,  are handled by  existing JSBACH

690 modules which were not modified in the context of the presented study. With an exclusive focus on simulating processes at site-level scale, it may even be possible to upgrade these modules and add some features that would be relevant for the methane processes

695 ; however, since our scope was to provide a methane extension for JSBACH that can be applied globally, certain limitations regarding the representation of site level observations need to be taken into account. This situation is even aggravated due to the use of parameter settings from global fields, i.e. with a coarse spatial resolution that aggregate conditions over larger areas and thus naturally cannot

700 provide the exact details for the field site where the reference fluxes were measured.  Such systematic deviations in modelling framework and parameter configuration will generate systematic differences between model output and site level measurements.  Accordingly, modelled hydrological con-

705 ditions and amounts of decomposed carbon need to be considered when comparing modelled methane fluxes to the site level observations and interpreting the spatiotemporal differences.

As mentioned above, the JSBACH hydrology module has been designed for global applications, and

710 is not capable to capture conditions in complex landscapes such as polygonal tundra. Therefore, for the Samoylov site which we used for this site level analysis, the modelled soil climate and hydrology systematically deviate from those found in the field (Beer, 2016) . We still chose to work at this site since a highly valuable interdisciplinary dataset could be provided to evaluate different facets of the model output. To adapt the model to represent the complex hydrology, a mixed approach

715 of combining two different model runs was applied. This approximation implies a very simplified representation of the real hydrological conditions and cannot fully offset all site level differences between model simulations and observational datasets. Accordingly, systematic biases need to be considered when interpreting the findings. However, through this approach we could demonstrate the paramount importance of realistic hydrologic boundary conditions for simulations of the methane

720 balance. In many aspects, details in the behaviour of the methane processes are tightly linked to the spatiotemporal variation of hydrological conditions, therefore biases in hydrology are directly projected on the methane processes.

Still, the  authors believe that the comparison of methane simulations against selected

725 site level measurements are an important first step to evaluate the overall performance of the new module.

It is obvious that the limitations of the observational database employed herein, i.e. using just one single observation site and focusing on the growing season alone, cannot allow for a comprehensive assessment of the newly implemented algorithms. Accordingly, the limited amount of available

730 field measurements from chamber and eddy covariance based fluxes requires a careful interpretation when compared to model results. , particularly regarding the evaluation of JSBACH as a process-oriented global biosphere model. For the Arctic domain, methane emissions during shoulder and winter seasons have been shown to add considerably to the full annual budget, an aspect that we

735 cannot evaluate based on the given database. Moreover, the question of temporal representativeness is complicated by the discontinuous nature of the methane fluxes  To overcome these limitations, in follow-up studies the authors plan to conduct model evaluations

740 based on longer-term flux measurements, covering full annual cycles for multiple Arctic sites.

Even  though we regard eddy covariance based fluxes  as the most reliable reference data source for longer-term site-level model evaluation, the influence of

745 microsite variability in the area surrounding the tower clearly poses a challenge here. Particularly with respect to methane fluxes, pronounced variability in the distribution of soil organic matter and water content may lead to a mosaic of different source strengths. For the Samoylov domain, which is characterised by polygonal structures, we mimicked the apparent differences between wet (center) and dry (rim) areas through the execution of two model runs with different settings. Still, the foot-

750 print composition of the eddy covariance tower might not match the mixed approach of 65 % rim and 35 % center used for modelling (Sachs et al., 2010).  Even though this mixture generally captures the composition of the larger area surrounding the tower, particularly when footprints are smaller during daytime  the

755 reduced field of view of the sensors might focus on areas that are wetter or drier than the average.

760  Our concept of combining two

is. For example, many details in the behaviour of the methane processes follow strictly the varying hydrological conditions during the year or between the microsites separate model runs has to be regarded as an approximation to cope with the hydrological constraints of a global model on the one hand, and the complex landscape on the other.

The model application for remote permafrost areas may also be limited by the availability of long-term and complete observations of meteorological data to be used as model forcing. Forcing data and methane fluxes are required for the same time period, which optimally lasts over one or more years. When going towards regional to global applications, this new model might be additionally compared to regional or global atmospheric inversion results (e.g. Bousquet et al., 2011; Berchet et al., 2015) or data-oriented data-driven upscaling of eddy covariance or chamber based observations (e.g. Christensen et al., 1995; Marushchak et al., 2015).

Within the methane module presented in this work, the discretisation as well as the definition of the pore volume are variable, thus . This requires that the time step of calculation and the diffusion coefficients must fit to the thicknesses of the soil layers. Otherwise If not set up properly, instabilities like oscillations or unrealistic behaviour like negative concentrations may occur. This However, since the new methane module has been designed to be flexible in this respect, and adjustments can easily be made in case numerical problems arise.

Furthermore, assumptions, e. g. about winter A parameter sensitivity study (section 3.4) shows that only for one parameter the uncertainty of the resulting methane emissions scales linearly with the uncertainty of the parameter. This parameter represents the amount of *in situ* (potential) methane produced under anaerobic conditions compared to the total *in situ* decomposition flux into carbon dioxide and methane $\left(\frac{[CH_4]}{([CO_2]+[CH_4])}\right)$. Based on the stoichiometry of the methanogenesis chemical reaction equation and based on laboratory and field data (Segers, 1998) , this parameter was chosen to be 0.5 in equation 2. In other models, this parameter is used as an effective parameter and has been tuned to match ultimate methane and carbon dioxide emissions from soil to the atmosphere in the absence of an explicit representation of oxygen and hence methanotrophy (Wania et al., 2010) .

Regarding our assumptions concerning fluxes or plant transport , might be too strict according to newer findings (Zona et al., 2016; Marushchak et al., 2015) . The prohibition of gas exchange with the atmosphere under conditions with more than during wintertime, according to recent findings (Zona et al., 2016; Marushchak et al., 2015) the settings chosen within the context of this manuscript might be oversimplifying the actual processes in the field. Our mechanism that prevents methane release once the snow cover reaches a depth of 5 of snow on the ground is an adaption to cm is a very crude approximation of the snow cover influence. It resulted from biases in the modelled

hydrological conditions in winter, where freezing of relatively dry soils led to oxic soil conditions that facilitated methane transport into the soil. The next iteration of the model development will include a more sophisticated, process-based representation of methane diffusion through snow. This upgrade, however, needs to be coupled to a major restructuring of several model components, and thus cannot be reconciled with the model version presented within the context of this manuscript.

The  implementation of the plant transport follows a mechanistic approach, but its definition is limited by the availability of observational evidence on e.g. diffusion velocities. Therefore, the parameter settings used in this study are subject to high uncertainty. The value for the diffusion coefficient in the exodermis was chosen to be 80 % of the diffusion coefficient in water  (*pers. comm.* C. Knoblauch). The subsequent gas transport within the aerenchyma is assumed to be as quick as diffusion in air.  With this setup, the effective barrier of the root exodermis  will limit the plant transport efficiency, and therefore act as a dominant control for this emission pathway. The thickness of this barrier has a large influence on plant transport  as well, i.e. a thinner root exodermis would lead to  increased plant transport. While this parameter is relatively easy to define, the cumulative surface area of all gas transporting roots in the soil column is ~~not at all easy to determine. But the larger this surface, the larger the plant transport . If it is found, that plant transport is too lowcompared to the other transportpathways, it is likely that also the chosen value for the surface area of gas transporting roots is not yet optimal. These kind of issues are the subject of ongoing investigations, but the module has been designed flexible, and adjusting of parameters~~difficult to constrain. Considering our basic assumption that plant transport is slower than diffusion in water, the general patterns of flux processes and soil moisture for rim and center conditions appear plausible. Regarding the quantitative flux rates, however, the fraction of the total flux emitted through plant transport in the model tends to be too low. With larger root surface leading to increased plant transport, we therefore could use this setting as a tuning parameter to improve this issue. But also the oxygen available to consume methane plays another modulating role, particularly for plant transport. Accordingly, new observational evidence would certainly improve the associated uncertainties, therefore this issue is subject to ongoing investigations. With the new methane module, designed to be flexible regarding these kind of settings, parameter adjustments with respect to newer findings can be easily implemented.

The contribution of labile root exudates to methane production and emission has been largely neglected in existing model implementations and is also not considered in this model configuration. This is also an understudied process in field experiments and can only be estimated indirectly. The rate of root exudates is linked to the nutrient availability in soils, with more root exudates present in plants located in nutrient poor wetland soils (Koelbener et al., 2010). The wetland soils in Arctic tundra are known to be nitrogen limited (Melle et al., 2015; Gurevitch et al., 2006). The plant growth in the polygonal lowland tundra of Indigirka, Russia, is co-limited by nitrogen and phosphorus and only about 5 % of the total nitrogen soil content is active in the biological fraction (Beerman et al., 2015). The presence of vascular plants in Arctic wetlands support the production of highly labile low molecular weight carbon compounds which can promote methane emissions through their methanogenic decomposition (Ström et al., 2012). An indirect evidence of the role of root exudates to methane production in polygonal ponds and water-saturated soils in Samoylov Island is presented by Knoblauch et al. (2015). The authors found almost 4 fold higher potential methane production rates in vegetated sites compared to the non-vegetated ones, both with the same C and N soil concentrations. Thus, the contribution to methane emissions from wetland soils in Arctic tundra due to the decomposition of root exudates should be taken into account in models. This will allow the understanding of the role of root exudates under present climate conditions. On the other hand, the potential nutrient mobilisation in soils due to permafrost degradation under climate change (Kuhry et al., 2010) may reduce the role of root exudates to methane emissions. However, the current JSBACH configuration lacks of a full soil nutrient cycle and the assimilation of nutrients by plant roots, as well as the contribution of root exudates to the total methane emissions cannot be modelled at this point.

In Samoylov Island, the minimum of modelled daily sums of methane emissions during summer is smaller and the maximum much higher for rim and center compared to measurements published by Kutzbach et al. (2004). However, these observations do not include spring bursts with very short but also very high emissions or even dry phases with small uptake. On the other hand, the mean of those measurements is 3 times as high for rim and 3.5 times as high for center compared to the modelled daily sums in summer (Table 3). Such high modelled emissions are rather rare when comparing previously published studies, where the general level of modelled values is lower than in observations (Fig. 7).

When comparing our model results at Samoylov Island to published results from other high-latitude regions, reasonable agreement is found. Our modelling results are about 40 to 60 % lower than measurements for BOREAS, Canada, and Abisko, Sweden, (Wania et al., 2010). The Lena River Delta region is much colder and drier compared to these sites, suggesting that lower flux rates are indeed reasonable. Furthermore, the Samoylov site is characterised by mineral soils containing substantially lower organic carbon as substrate for methane

production than the organic soils at the BOREAS site and the mire in Abisko. Compared to measurements done by Desyatkin et al. (2009) on a thermokarst terrain at the Lena river near Yakutsk, our mean results are well within the measurement range if comparing our rim to the drier sites, our center to the wetter sites, and our mixed approach to the entire ecosystem (Table 4).  However, climate and environmental conditions  in this study were very different from those in  observed in Samoylov, thus this comparison can only be regarded as a rough guideline. Nakano et al. (2000) measured methane fluxes at Tiksi near the mouth of the Lena river. While our mean value at rim is 4.5 times as high as the mean measurements in Tiksi, the mean at the center is 5.5 times as high as our mean value (Table 3). The modelled minimum is lower for the center but comparable for the rim.

The large methane spring burst simulated by the model in both rim and center  may represent the release of methane that has been accumulated during winter in the topsoil below the snow layer. To our knowledge, there is no observational reference of spring bursts  measured in Samoylov Island. However, evidence of these events have been presented for other wetland areas using chambers and eddy covariance measurements, e.g. in north Sweden, Jammet et al. (2015) and Friborg et al. (1997) ; in Finland, Hargreaves et al. (2001) ; in north Japan, Tokida et al. (2007b) and in Northeast China, Song et al. (2012) . These studies suggest the presence of spring thaw emissions of methane that occur sporadically over short periods in the form of bursts. The magnitude of the spring bursts can exceed the mean summer fluxes by a factor of 2 to 3. Although spring emissions can account for a large share of the total annual fluxes, their occurrence, duration and magnitude are still uncertain. To adequately characterise the spring bursts in Samoylov Island, it is necessary to perform dedicated field measurements during the spring thaw period. These results will then help to evaluate the representativeness of the modelled spring bursts. In future model iterations, the spring bursts will also be evaluated for larger spatial scales.

 Zona et al. (2009), several measurements of methane emissions in the Arctic tundra are  presented. Despite our mean values are located towards the lower end, our minimum, mean and maximum values fit well within the given range. Bartlett et al. (1992) measured methane fluxes near Bethel in the Yukon–Kuskokwim delta, Alaska. The provided values for upland tundra compare well to our mean and minimum values. However, the model maximum fluxes are higher than the measurement values for upland tundra but still well in the range of measured values for wet meadow, which has higher moisture contents than upland tundra. In fact, the highest values are calculated if soil moisture is highest, so despite more on the lower end of this water logged landscape type's emissions, they fit well also therein.  Summarising, the

variability of results of this pan-Arctic survey indicates that methane budgets within all these places  are influenced by different conditions in terms of weather,  hydrology and carbon pools. Accordingly, the good agreement of our modelled

915  values with these references confirm that our results are within a plausible range at the greater picture, but a detailed evaluation cannot be performed without in-depth analysis of the site-level conditions.

Regarding the general structure of the JSBACH model, other parts of the land surface scheme
920  require advancements before applying it with the methane module at global scale and over long time periods can be suggested. For example, soil organic matter should be represented vertically resolved (Braakhekke et al., 2011, 2014; Koven et al., 2015; Beer, 2016) , with different soil carbon pools and a moisture dependent decomposition. Furthermore, the site hydrology should include  water contents above field capacity,
925  and standing water  above the surface (Stacke and Hagemann, 2012) . We are also aware, however, that it is not the best approach to calculate an empirical water table depth  after Stieglitz et al. (1997) from  unsaturated soil water  conditions. Together with the water table depth, the soil moisture content itself is of great importance to the presented methane module.
930   Still, with this model version, the importance of different processes, their interplay  and the influence of climatic or hydrologic drivers can be studied at site level, which is a major step forward. Furthermore, this process-based implementation can be applied at other sites or with another hydrology, and still, the methane-related processes will only depend on the soil conditions. In order to improve the hydrological scheme of the current model version, it
935  would be desirably to use other approaches like TOPMODEL (e.g. Kleinen et al., 2012) that would allow representing the fraction of the inundated area in a model grid cell based on the topography profile. This would provide a modelled wetland extent and a representation of the water table depth in saturated soils, especially for large-scale applications. This step is been considered and will be included in future model iterations. 
[revised manuscript text omitted]

Beer, C.: Permafrost Sub-grid Heterogeneity of Soil Properties Key for 3-D Soil Processes and Future Climate Projections, Front. Earth Sci., 4, 1–81, doi:10.3389/feart.2016.00081, 2016.

Beerman, F., Teltewskoi, A., Fiencke, C., Pfeiffer, E.-M., Kutzbach, L.: Stoichiometric analysis of nutrient availability (N,P,K) within soils of polygonal tundra, Biogeosciences, 122, 2, 211–227, doi:10.1007/s10533-014-0037-4, 2015.

Berchet, A., Pison, I., Chevallier, F., Paris, J.-D., Bousquet, P., Bonne, J.-L., Arshinov, M. Y., Belan, B. D., Cressot, C., Davydov, D. K., Dlugokencky, E. J., Fofonov, A. V., Galanin, A., Lavrič, J., Machida, T., Parker, R., Sasakawa, M., Spahni, R., Stocker, B. D., Winderlich, J.: Natural and anthropogenic methane fluxes in Eurasia: a mesoscale quantification by generalized atmospheric inversion, Biogeosciences, 12, 18, 5393–5414, doi:10.5194/bg-12-5393-2015, 2015.

Boike, J., Kattenstroth, B., Abramova, K., Bornemann, N., Chetverova, A., Fedorova, I., Fröb, K., Grigoriev, M., Grüber, M., Kutzbach, L., Langer, M., Minke, M., Muster, S., Piel, K., Pfeiffer, E.-M., Stoof, G., Westermann, S., Wischnewski, K., Wille, C., Hubberten, H.-W.: Baseline characteristics of climate, permafrost and land cover from a new permafrost observatory in the Lena River Delta, Siberia (1998-2011), Biogeosciences, 10, 3, 2105–2128, doi:10.5194/bg-10-2105-2013, 2013.

Boudreau, B. P.: Diagenetic Models and their Implementation. Modelling Transport and Reactions in Aquatic Sediments, Springer, Berlin, doi:10.1007/978-3-642-60421-8, 1997.

Bousquet, P., Ringeval, B., Pison, I., Dlugokencky, E. J., Brunke, E.-G., Carouge, C., Chevallier, F., Fortems-Cheiney, A., Frankenberg, C., Hauglustaine, D. A., Krummel, P. B., Langenfelds, R. L., Ramonet, M., Schmidt, M., Steele, L. P., Szopa, S., Yver, C., Viovy, N., Ciais, P.: Source attribution of the changes in atmospheric methane for 2006–2008, Atmos. Chem. Phys., 11, 8, 3689–3700, doi:10.5194/acp-11-3689-2011, 2011.

Braakhekke, M. C., Beer, C., Hoosbeek, M. R., Reichstein, M., Kruijt, B., Schrumpf, M., Kabat, P.: SOMPROF: A vertically explicit soil organic matter model, Ecological Modelling, 222, 10, 1712–1730, doi:10.1016/j.ecolmodel.2011.02.015, 2011.

Braakhekke, M. C., Beer, C., Schrumpf, M., Ekici, A., Ahrens, B., Hoosbeek, M. R., Kruijt, B., Kabat, P., Reichstein, M.: The use of radiocarbon to constrain current and future soil organic matter turnover and transport in a temperate forest, J. Geophys. Res. Biogeosci., 119, 3, 372–391, doi:10.1002/2013JG002420, 2014.

Christensen, T. R., Jonasson, S., Callaghan, T. V., Havström, M.: Spatial variation in high-latitude methane flux along a transect across Siberian and European tundra environments, J. Geophys. Res.: Atmospheres (1984–2012), 100, D10, 21035–21045, doi:10.1029/95JD02145, 1995.

Christensen, T. R., Prentice, I. C., Kaplan, J., Haxeltine, A., Sitch, S.: Methane flux from northern wetlands and tundra. An ecosystem source modelling approach, Tellus B, 48, 5, 652–661, doi:10.1034/j.1600-0889.1996.t01-4-00004.x, 1996.

Collin, M., Rasmuson, A.: A comparison of gas diffusivity models for unsaturated porous media, Soil Sci. Soc. Am. J., 52, 6, 1559–1565, doi:10.2136/sssaj1988.03615995005200060007x, 1988.

Dean, J. A.: Lange's Handbook of Chemistry, McGraw-Hill, Inc., ISBN 0-07-016384-7, 1992.

Desyatkin, A. R., Takakai, F., Fedorov, P. P., Nikolaeva, M. C., Desyatkin, R. V., Hatano, R.: CH4 emission from different stages of thermokarst formation in Central Yakutia, East Siberia, Soil Science Plant Nutr., 55, 4, 558–570, doi:10.1111/j.1747-0765.2009.00389.x, 2009.

Ekici, A., Beer, C., Hagemann, S., Boike, J., Langer, M., Hauck, C.: Simulating high-latitude permafrost regions by the JSBACH terrestrial ecosystem model, Geosci. Model Dev., 7, 2, 631–647, doi:10.5194/gmd-7-631-2014, 2014.

Friborg, T., Christensen, T. R., Sogaard, H.: Rapid response of greenhouse gas emission to early thaw in a subarctic mire as shown by micrometeorological techniques, Geophys. Res. Lett., 24, 23, 3061–3064, doi:10.1029/97GL03024, 1997.

Gurevitch, J., Scheiner, S. M., Fox, G. A.: The Ecology of Plants, 2nd ed., Sinauer Associates, Inc., Sunderland, ISBN: 0878932941, 2006.

Hagemann, S., Stacke, T.: Impact of the soil hydrology scheme on simulated soil moisture memory, Clim. Dyn., 44, 7, 1731–1750, doi:10.1007/s00382-014-2221-6, 2014.

Harden, J. W., Koven, C. D., Ping, C.-L., Hugelius, G., McGuire, A. D., Camill, P., Jorgenson, T., Kuhry, P., Michaelson, G. J., O'Donnell, J. A., Schuur, E. A. G., Tarnocai, C., Johnson, K., Grosse, G.: Field information links permafrost carbon to physical vulnerabilities of thawing, Geophys. Res. Lett., 39, 15, L15704, 1–6, doi:10.1029/2012GL051958, 2012.

Hargreaves, K. J., Fowler, D., Pitcairn, C. E. R., Aurela, M.: Annual methane emission from Finnish mires estimated from eddy covariance campaign measurements, Theor. Appl. Climatol., 70, 1, 203–213, doi:10.1007/s007040170015, 2001.

Hugelius, G., Strauss, J., Zubrzycki, S., Harden, J. W., Schuur, E. A. G., Ping, C.-L., Schirrmeister, L., Grosse, G., Michaelson, G. J., Koven, C. D., O'Donnell, J. A., Elberling, B., Mishra, U., Camill, P., Yu, Z., Palmtag, J., Kuhry, P.: Estimated stocks of circumpolar permafrost carbon with quantified uncertainty ranges and identified data gaps, Biogeosciences, 11, 23, 6573–6593, doi:10.5194/bg-11-6573-2014, 2014.

Ito, A., Inatomi, M.: Use of a process-based model for assessing the methane budgets of global terrestrial ecosystems and evaluation of uncertainty, Biogeosciences, 9, 2, 759–773, doi:10.5194/bg-9-759-2012, 2012.

Jackowicz-Korczyński, M., Christensen, T. R., Bäckstrand, K., Crill, P., Friborg, T., Mastepanov, M., Ström, L.: Annual cycle of methane emission from a subarctic peatland, J. Geophys. Res., 115, G2, G02009, 1–10, doi:10.1029/2008JG000913, 2010.

Jähne, B., Heinz, G., Dietrich, W.: Measurement of the diffusion coefficients of sparingly soluble gases in water, J. Geophys. Res., 92, C10, 10767–10776, doi:10.1029/JC092iC10p10767, 1987.

Jammet, M., Crill, P., Dengel, S., Friborg, T.: Large methane emissions from a subarctic lake during spring thaw: Mechanisms and landscape significance, J. Geophys. Res. Biogeosci., 120, 11, 2289–2305, doi:10.1002/2015JG003137, 2015.

Jansson, P.-E., Karlberg, L.: COUP Manual. Coupled heat and mass transfer model for soil-plant-atmosphere systems, www.coupmodel.com, 2011.

Khvorostyanov, D. V., Krinner, G., Ciais, P., Heimann, M., Zimov, S. A.: Vulnerability of permafrost carbon to global warming. Part I: model description and role of heat generated by organic matter decomposition, Tellus B, 60, 2, 250–264, doi:10.1111/j.1600-0889.2007.00333.x, 2008.

Kleinen, T., Brovkin, V., Schuldt, R. J.: A dynamic model of wetland extent and peat accumulation: results for the Holocene, Biogeosciences, 9, 1, 235–248, doi:10.5194/bg-9-235-2012, 2012.

Knoblauch, C., Spott, O., Evgrafova, S., Kutzbach, L., Pfeiffer, E.-M.: Regulation of methane production, oxidation, and emission by vascular plants and bryophytes in ponds of the northeast Siberian polygonal tundra, J. Geophys. Res., 120, 12, 2525–2541, doi:10.1002/2015JG003053, 2015.

Koelbener, A., Ström, L., Edwards, P. J., Venterink, H. O.: Plant species from mesotrophic wetlands cause relatively high methane emissions from peat soils, Plant Soil, 326, 1, 147–158, doi:10.1007/s11104-009-9989-x, 2010.

[revised manuscript text omitted]

Maximal values of the cumulative sums of modelled methane flux over the modelled time period for rim, center and a mixed approach of 65 % rim plus 35 % center for the different transport processes and combined in $\mathrm{gC\,m^{-2}}$, rounded to three non-zero digits.

**Table 2.** Methane emission sensitivity towards key parameter settings.

| Parameter                | lower range | upper range |
| ------------------------ | ----------- | ----------- |
| fracCh4Anox              | -11.966     | 12.035      |
| fracO2forOx+fracO2forPh  | -1.358      | 1.305       |
| KmO2                     | -1.741      | 2.107       |
| snowThresh               | 0.549       | -0.090      |
| resistRoot               | 0.024       | 0.195       |
| thickExoderm             | 0.204       | 0.032       |
| rootLength               | 0.024       | 0.195       |
| rootDiam                 | 0.024       | 0.195       |
| tillerNumberMax          | 0.024       | 0.195       |
| dominanceCarexAquatilis  | -0.151      | 0.344       |

Percentage change of the cumulative methane emissions over the modelled time period, when the parameter was modified by +/-10 %, compared to its default setting.

[revised manuscript text omitted]